# Janus particles with tunable patch symmetry and their assembly into chiral colloidal clusters

Tianran Zhang[1,3], Dengping Lyu [1,3], Wei Xu[1,3], Xuan Feng[2], Ran Ni [2] ✉ & Yufeng Wang [1] ✉

Janus particles, which have an attractive patch on the otherwise repulsive surface, have been commonly employed for anisotropic colloidal assembly. While current methods of particle synthesis allow for control over the patch size, they are generally limited to producing dome-shaped patches with a high symmetry ($C_\infty$). Here, we report on the synthesis of Janus particles with patches of various tunable shapes, having reduced symmetries ranging from $C_{2v}$ to $C_{3v}$ and $C_{4v}$. The Janus particles are synthesized by partial encapsulation of an octahedral metal-organic framework particle (UiO-66) in a polymer matrix. The extent of encapsulation is precisely regulated by a stepwise, asymmetric dewetting process that exposes selected facets of the UiO-66 particle. With depletion interaction, the Janus particles spontaneously assemble into colloidal clusters reflecting the particles' shapes and patch symmetries. We observe the formation of chiral structures, whereby chirality emerges from achiral building blocks. With the ability to encode symmetry and directional bonding information, our strategy could give access to more complex colloidal superstructures through assembly.

Colloidal superstructures by spontaneous assembly of nano- and microscale particles hold promise for applications such as photonics[1], catalysis[2], sensing[3], and liquid transport[4]. Janus particles, made by selectively modifying a patch or hemisphere of an otherwise isotropic particle, enable directional colloidal interactions and are crucial building blocks for assembling complex superstructures. Many studies, including both experiments and simulations, have investigated Janus particles of different types, and their assembly has led to the formation of colloidal clusters[1,5–7], rings[1], fibers[1,8–11], etc. In most cases, however, adjusting the size of the patches is the primary handle to tune the assembly results, which limits the variety of colloidal structures one may obtain[12]. Strategies for further control of the bonding of Janus particles are urgently needed.

Recently, it has been suggested that altering the shape of the patch could lead to a different outcome for Janus particles in terms of their interaction and assembly[1,13,14]. Indeed, most previous Janus particles have a high-symmetry ($C_\infty$) dome-shaped patch (i.e., hemisphere or a spherical cap), which is derived from the spherical precursor particles such as polymer latices[1,5,13,15,16] and colloidal silica[17–21]. A patch with a reduced symmetry would, in principle, contain more anisotropic information and therefore impose increased directionality for particle assembly[22]. So far, only a few works have mentioned Janus particles with patches that deviate from the dome shape. For example, Janus particles with a pentagonal patch have been observed for particles made by a mix-and-melt method[22,23]; faceted or patterned patches have also been demonstrated[14,24,25]. However, there is no general method for producing Janus particles with fidelity whose patch shape and symmetry can be finely controlled and systematically tuned. Consequently, it remains unexplored how a shaped patch can enable different colloidal superstructures through assembly.

[1]Department of Chemistry, The University of Hong Kong, Pokfulam Road, Hong Kong SAR, China. [2]School of Chemistry, Chemical Engineering and Biotechnology, Nanyang Technological University, Singapore, Singapore. [3]These authors contributed equally: Tianran Zhang, Dengping Lyu, Wei Xu. ✉e-mail: r.ni@ntu.edu.sg; wanglab@hku.hk

In this work, we report on a distinct type of Janus particles that possess synthetically tunable patch shapes and symmetries. We achieve this by partially embedding an octahedral metal-organic framework (MOF) particle into a polymerizable oil matrix. The extent of encapsulation is systematically adjusted by a surfactant-aided, stepwise dewetting process, which selectively exposes one or multiple facets of the MOF particle and determines the relative particle orientation. The synthesis has a high yield, does not require tedious purifications, and can generate particles/patches adopting $C_{2v}$ to $C_{3v}$, and $C_{4v}$ symmetries. Simulation validates that the particle shapes are a result of minimization of surface energy. When we introduce depletion interaction, the Janus particles spontaneously assemble with site-selectivity to form colloidal clusters reflecting the Janus particles' characteristics. Notably, we observe the formation of chiral colloidal assemblies, whereby chirality emerges from achiral building blocks. The assembly results have also been verified by Monte Carlo simulation.

## Results

### Synthesis of Janus particles with various patch symmetry

MOFs are porous crystalline materials composed of metal nodes and organic linkers[26]. MOFs can be synthesized as nano- and microscale crystals with polyhedral shapes, which, as building blocks, have been recently assembled into various colloidal superstructures[27–30]. In this paper, we exploit colloidal MOF particles to construct our Janus particles.

The synthesis, as shown in Fig. 1a, begins with heterogeneous nucleation of an oil droplet on a solid MOF particle. The oil is derived by hydrolyzing and condensing 3-(trimethoxysilyl)propyl methacrylate, TPM, while the MOF particles are microcrystals of the zirconium-based UiO-66 framework with an octahedral shape. Typically, the particle size, measuring the edge length of the octahedron $\varphi$, range from 500 nm to 1.5 μm, with a narrow size distribution (see Supplementary Figs. 1, 2, and Supplementary Note 1). The nucleation of TPM

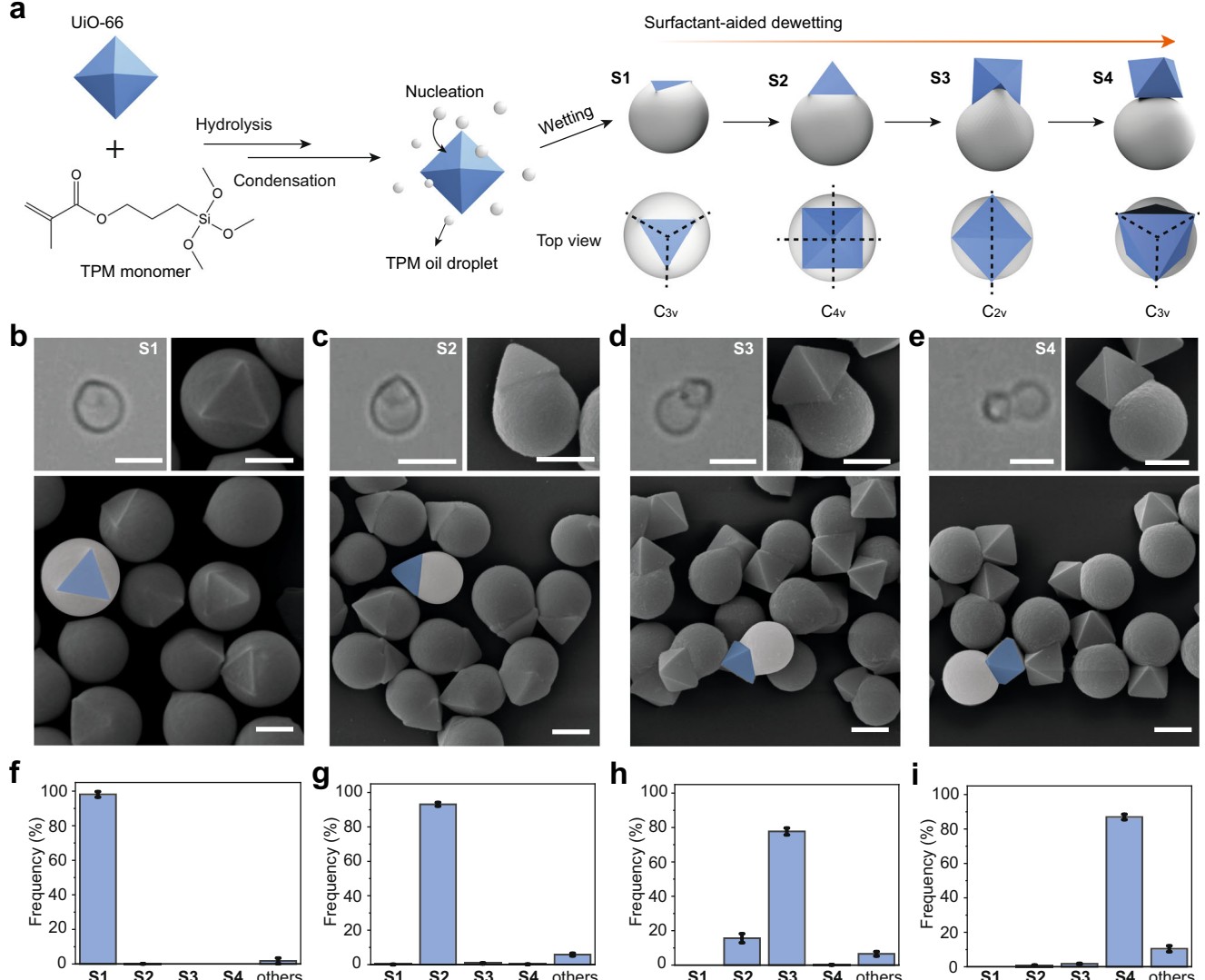

**Fig. 1 | Synthesis of Janus particles with various patch shapes/symmetries. a** Schematic showing the encapsulation (wetting) and dewetting strategy to synthesize Janus particles with various symmetries. In basic solution, TPM monomer undergoes hydrolysis and condensation, from which TPM oil droplets are produced (or nucleated). Octahedral UiO-66 microcrystals (shown in blue) are encapsulated by the surrounding TPM oil droplet (shown in gray) nucleated on its surface, result in the wetting of the microcrystals by the oil droplets. In the presence of Triton X-100, the oil droplets are dewetted from the surface of UiO-66

particles, leading to the Janus particles with different symmetries. Optical bright-field images (top-left) and scanning electron microscope (SEM) images of single (top-right) and multiple Janus particles (bottom) of **S1** (**b**), **S2** (**c**), **S3** (**d**), and **S4** (**e**). Number frequency of the as-synthesized Janus particles of **S1** (**f**), **S2** (**g**), **S3** (**h**), and **S4** (**i**). Error bars are standard deviations. The measurement for frequency is repeated 3 times and at least hundreds of particles are measured each time. Scale bars: 1 μm for SEM images and 2 μm for optical bright-field images.

oil on the UiO-66 particle is presumably due to their oppositely charged surfaces (zeta potential $\zeta_{UiO\text{-}66}$ = +23 mV and $\zeta_{TPM}$ = -30 mV). The process leads to hybrid particles with the UiO-66 particle almost fully engulfed in the TPM oil droplet.

A surfactant-aided dewetting strategy[31] is then implemented to tune the morphology of the hybrid particles. A nonionic surfactant, Triton X-100 (TX), is employed, which adsorbs on the surfaces of TPM and UiO-66 with different affinity. It is hypothesized that UiO-66 absorbs more TX on its surface than TPM does, resulting in a more rapid decrease of surface tension between UiO-66 and water than that between TPM and water, which increases the contact angle between TPM and UiO-66. The process causes the TPM oil to dewet and expose the UiO-66 particle, resulting in Janus particles with a shaped patch. By tuning the concentration of TX, four representative particle configurations, S1–S4, have been observed, each with a Janus shape (or a hybrid dimer shape) but distinct patch symmetries (Fig. 1a). Solidifying the TPM oil by thermal or photo-polymerization fixes the particle configurations.

The shape of S1 particles, obtained when no TX is added, is nearly spherical. Scanning electron microscope (SEM) images show that the UiO-66 particle is encapsulated almost within the TPM oil. However, one of its triangular facets is clearly exposed, which becomes a triangular patch signifying a $C_{3v}$ symmetry (Fig. 1b). With TX added, at 0.33 mM, S2 particles are formed, and the exposed UiO-66 part (or the patch) is a square pyramid displaying four facets, having a $C_{4v}$ symmetry (Fig. 1c). With more TX (0.66 mM), only two adjacent facets of the UiO-66 particle are left in contact with the TPM lobe (Fig. 1d). This affords the S3 particles with a $C_{2v}$ symmetry. Increasing the concentration of TX to 1.32 mM leads to the fully dewetted S4 particles, where the TPM oil droplet only attaches to one facet of UiO-66 particle. The particle exhibits a $C_{3v}$ symmetry based on the UiO-66 patch (Fig. 1e). The TPM oil detaches UiO-66 particle upon further increase of TX.

In all cases, and regardless of the size of the MOF particles tested, the obtained Janus particles have consistent morphologies, featuring high uniformity and purity (Fig. 1f–i and Supplementary Fig. 3). The size dispersity is typically below 0.1 (Supplementary Fig. 2 and Supplementary Note 1). Also, the dewetting process is instant and the transition between different particle configurations can be monitored in situ by optical microscopy. For example, Supplementary Movie 1 shows the process of S2 being converted to S3.

## Surface Evolver simulation

One feature of our synthesis is that, as dewetting occurs, the exposed part of the UiO-66 particle (i.e., the patch) adopts different symmetries. This contrasts the dewetting of a droplet from a spherical particle, which generates dumbbell Janus particles with a dome-shaped patch, having a fixed $C_{\infty}$ symmetry (Fig. 2a). Moreover, for MOF-based Janus particles, only discrete configurations with characteristic shapes (namely S1–S4) are obtained, yet for dumbbell Janus particles, continuous dewetting can in principle lead to an infinite number of shapes (of a dumbbell) that differ only in the particle's aspect ratio.

Because dewetting happens quickly upon the addition of surfactant, the final particle configuration is a result of surface energy minimization. We thus use Surface Evolver simulation[32] to rationalize the dewetting process and the particles shapes (Fig. 2). Models are built to compare different scenarios.

For dewetting from the surface of a spherical particle, simulation shows that the droplet moves outwards to expose the spherical particle until a set contact angle ($\theta$) is reached (Fig. 2a, Supplementary Movie 2). When $\theta$ is increased, the contact area continuously decreases, while the moving direction of the droplet remains the same (indicated by red arrows, referred to as symmetric dewetting or unidirectional dewetting) (Fig. 2a, d). However, for octahedral UiO-66

particle, the change in the contact area upon TPM dewetting is stepwise (Fig. 2b, d). We note that the shape evolution is only possible by asymmetric dewetting, whereby the oil droplet moves towards changing directions, presumably influenced by the sharp edges of the octahedron (Fig. 2b). The corresponding shapes of Janus particles agree with the various configurations (S1–S4) obtained in our experiments. Through simulation, we have identified the range of $\theta$ values for each particle configuration (Fig. 2d). For S1, $\theta \leq 55°$, while $60° \leq \theta \leq 80°$, S2 is obtained. For realizing S3 and S4, the contact angle should be $85° \leq \theta \leq 95°$ and $\theta \geq 100°$, respectively.

To explore the origin of the stepwise asymmetric dewetting, we set up a control group with additional constraints to restrict the direction of oil movement (Fig. 2c). Comparatively, this symmetric/unidirectional dewetting is associated with a higher surface energy for a wide range of contact angles ($40° < \theta < 100°$) and is therefore energetically unfavored (Fig. 2e). Interestingly, for asymmetric dewetting, we find that the ratio between the relative contact area and the number of facets in contact with oil ($\chi$) stays roughly unchanged during the evolution (Fig. 2f). This means that the oil droplet could either establish a good contact with one or more facets of the polyhedron, or it does not contact them at all (Supplementary Movie 3). In other words, the droplet tends to avoid crossing the edges of the octahedron at the liquid front. Above all, the polyhedral shape is the cause of these unexpected observations.

When $\theta$ reaches above 100°, the surface energies of both symmetric and asymmetric particles converge; the contact area is already smaller than the single facet of UiO-66 (Fig. 2e).

## Strategy for self-assembly

Our Janus particles are hybrid dimers with a spherical lobe and a multi-faceted patch. To enable their assembly while exploiting their shape characteristic, we set to utilize depletion force. As a short-ranged attraction induced by small depletants (e.g., non-adsorbing nanoparticles), depletion interaction has been previously employed in colloidal assembly. It is worth noting that in such assemblies, site-selectivity has often been observed for particles with different charges[33], surface roughness[6], and shapes[34].

We use micelles of cetyltrimethylammonium chloride (CTAC) as the depletant for depletion attractive force[35,36]. Prior to assembling the Janus particle, the suitable concentration of CTAC is determined by investigating a model system containing pure UiO-66 octahedra and TPM spheres (1 μm, about the size of the TPM lobe of Janus particle). The TPM spheres remain dispersed until CTAC reaches 7.5 mM, while the UiO-66 particles start to assemble and crystallize at 3 mM of CTAC resulting in hexagonal crystals (Fig. 3a, Supplementary Fig. 4). Therefore, within a wide window of CTAC concentration (3–7.5 mM), the assembly is selective towards the UiO-66 particles. Figure 3b are microscope images showing that, in a mixture, the UiO-66 particles form crystals (highlighted with blue trace) while the TPM particles are dispersed (fluorescent).

Among the various colloidal interactions, the depletion force is strongly shape-dependent and dominates the assembly. Specifically, the potential of depletion force, $U_{dp}$, can be approximated as $U_{dp} = -\Delta\Pi \times V_{OV}$ ($\Delta\Pi$) is the osmotic pressure change and $V_{OV}$ is the overlap volume between contacting particles, where the shape greatly influences the $V_{OV}$ value. Taking this into consideration, we calculate the overall interparticle potentials $U$ ($U = U_{dp} + U_{el} + U_{vdw}$, $U_{el}$ is electrostatic and $U_{vdw}$ is van der Waals interaction) between octahedral UiO-66 particles by their facets (facet/facet), between TPM spheres (sphere/sphere), and between UiO-66/TPM (facet/sphere) (Fig. 3c). The attraction between two planar facets (in the case of UiO-66) is most favored, presumably due to their large overlap volume and strong depletion force. These different binding scenarios can be found in systems of Janus TPM-UiO-66 particles and binding via the UiO-66 patches should prevail (Fig. 3d).

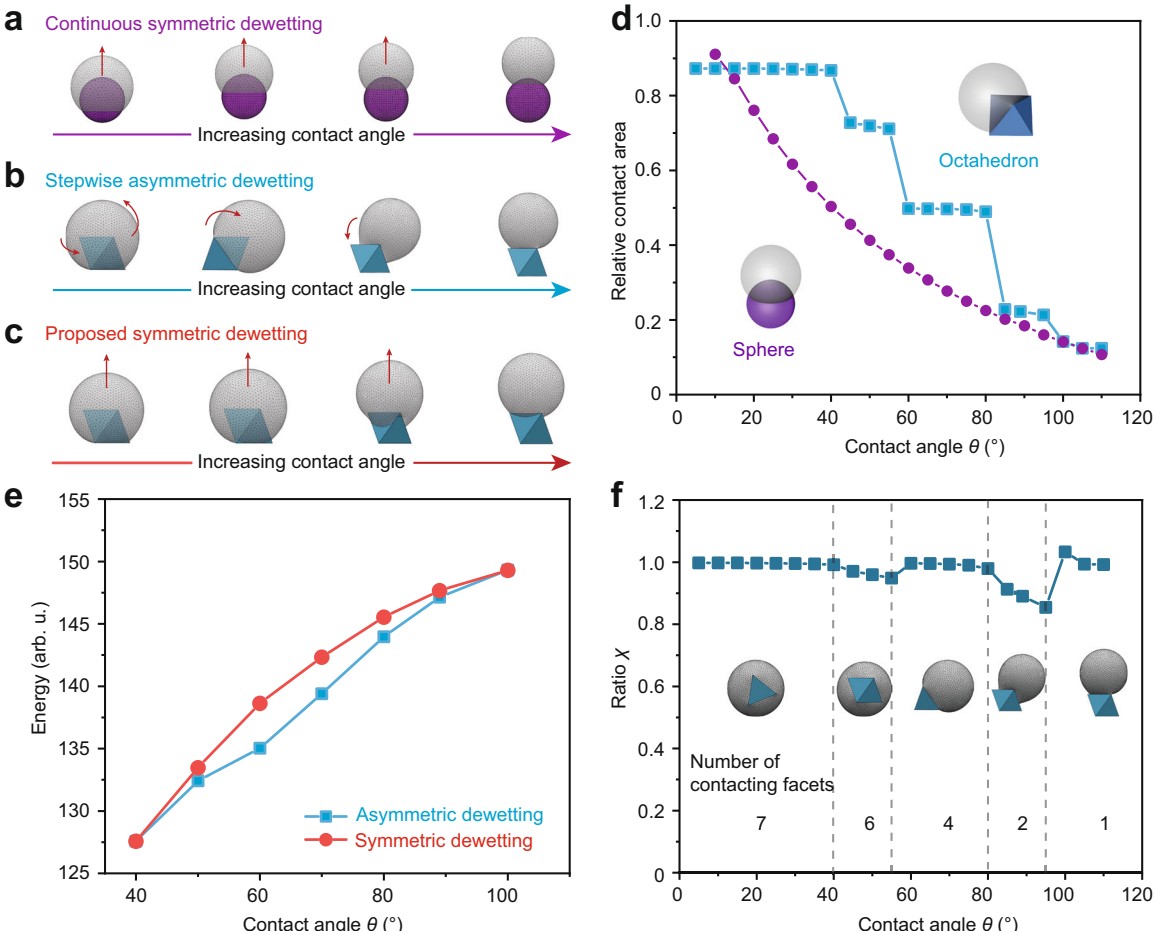

**Fig. 2 | Surface Evolver simulation. a–c** Illustration and results for different evolution scenarios. **a** Oil dewetting from a spherical particle (shown in purple) to form of dome-shaped Janus particles. The particles always adopt a dumbbell shape and a fixed $C_\infty$ symmetry. **b** Oil dewetting from an octahedral particle (shown in blue) to form Janus particles with various $C_n$ symmetries; the process is stepwise and asymmetric. **c** Oil dewetting an octahedron with a symmetric constraint (shown in red). The dewetting direction (or the direction of oil movement) is indicated by red arrows. **d** Plot showing the relative contact area (with respect to the overall surface of the particle) decreases continuously (for dewetting spherical particle) or stepwise (for dewetting octahedral particle), with increasing contact angle. **e** Comparison of the surface energies of Janus UiO-66 particles by asymmetric and symmetric dewetting. **f** The ratio $\chi$ between the relative contact area and the number of facets in contact stays roughly the same.

## Self-assembly of Janus particles (S1 and S2)

We next explore the self-assembly of Janus particles. By using 4 mM of CTAC, we expect that the UiO-66 patch can bind one another while the TPM lobes do not. Due to gravity, all the assembly occurs in 2D on a substrate, and the depletion attraction between particle facets and the flat substrate also plays a key role in determining the particle orientation and assembly.

For **S1** particles, they have only one UiO-66 facet exposed. With depletion force, the particle adheres to the substrate via a strong facet-to-substrate attraction (Supplementary Fig. 5). The facet facing the substrate can be characterized by confocal reflected microscopy, which clearly reveals a triangular pattern. Since no other facets are available for further interparticle binding, no assemblies are produced.

Unlike **S1**, the self-assembly of **S2** particles produces colloidal clusters in a high yield (Fig. 4). Most dominantly, the clusters are trimers in a trefoil shape, adopting a structure approximate to $C_{3v}$ symmetry ($\approx C_{3v}$) (Fig. 4a, b). Confocal microscopy indicates that, within a cluster, each Janus particle has one of their facets attached to the substrate (Supplementary Fig. 6). Since **S2** particles have four equivalent facets (with respect to the TPM lobe), there is only one orientation for them to sit on the substrate. By establishing interparticle contacts using other facets, the clusters are assembled. From confocal images (Fig. 4c), the relative position and orientation of each

particle can also be deduced, whereby the facets of two particles only slightly overlap to form a colloidal bond. Presumably, the TPM lobe provides a steric hindrance that impedes the further increase in contact area between the particle facets; the TPM also limits the number of particles per cluster, to be three in this case.

There are other minor assemblies of **S2**, namely dimers and Y-shaped trimers, both having a symmetry of $C_s$ (Fig. 4d, e). They are either the incomplete assembly product or structural isomer of the trefoil trimer. Having three colloidal bonds, the trefoil trimer is most thermodynamically stable species. While for the Y-trimers, only two bonds are formed (Fig. 4f).

## Assembly of chiral clusters from S3 and S4

For **S3** particles, given their shape, there are two types of inequivalent facets, labelled as α and β (Fig. 5a–c). There are four α facets next to the wetting facets and two β facets that are further away (Fig. 5a, d). The particle may adopt two possible orientations when attaching to the substrate by depletion interaction. However, because of the steric hindrance of the TPM lobe, the facet α cannot establish a full facet-substrate contact (Fig. 5c). The particles therefore always sit on the substrate by its β facet (Fig. 5b).

Upon assembly, the **S3** particles predominantly form trefoil trimer clusters with symmetry of $C_3$, which in this case are chiral (Fig. 5f, g).

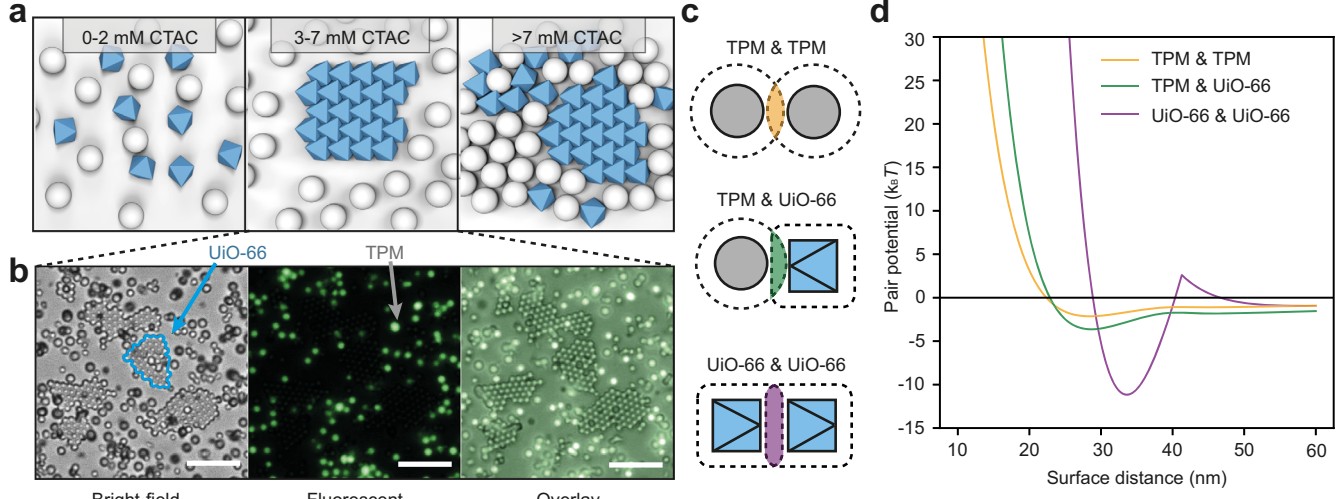

**Fig. 3 | Depletion interaction and shape selectivity. a** Cartoon showing the phase results of TPM and UiO-66 particles at different concentration of CTAC, which induces depletion force between particles. When CTAC is at 3–7 mM, the UiO-66 particles crystallize while the TPM particles disperse. **b** Bright-field (left), fluorescent (middle) and overlay (right) image showing the mixture of UiO-66 particles (no fluorescence) and TPM particles (green fluorescence) in 3 mM CTAC. **c** Cartoon illustration of the overlap of excluded volume between TPM particles (top), between TPM and UiO-66 particles (middle), and between UiO-66 particles (bottom). **d** Pairwise potential between TPM particles (yellow), between TPM and UiO-66 particles (green), and between UiO-66 particles (purple) in 3 mM CTAC. Scale bar: 20 μm for (**b**).

The assembly process is illustrated in Supplementary Movie 4, whereby the particles collide with one another; a dimer can form first by particles sticking by their UiO-66 patch, which is joined by a third particle to form the trimers. The long axes (or the $C_2$ symmetry axes) of the particles within a cluster do not converge to a point, signifying the chirality of the cluster (Fig. 5f). Clusters in both right- and left-handedness are observed in equal amount. From confocal images in Fig. 5e, g, the positions of the substrate-bound facets are shown, indicating the particle's bonding situation. There, the triangular facets of UiO-66 establish contact antiparallelly (Supplementary Fig. 7). The contact is no longer compromised by the TPM lobe, and the contacting area is maximized to have a stronger depletion interaction. While the UiO-66 parts are bonded within a cluster, the TPM lobes prevent the formation of larger, extended structures.

There are three minor species of **S3** assemblies, dimer (Fig. 5h), Y-shaped trimer (Fig. 5i), and tetramer (Fig. 5j), which coexist with the trefoil trimer. They possess symmetries of $C_1$, $C_s$ and $C_s$, respectively. Interestingly, the dimers are also chiral, while the other two species are not. The chirality of the dimers also comes from the arrangement of their $C_2$ axes, which have an angle of 120 degree without converging to a central point (Supplementary Fig. 8). For Y-shaped trimer and tetramer, mirror symmetries are found.

The trimer is the dominant species, and the chiral trefoil trimer outnumbers the achiral Y-shaped trimer (Fig. 5k). This is because of the maximum contact area (6 facets in contact) of trefoil trimer compared to that of Y-shaped trimer (4 facets in contact) and dimer (2 facets in contact). The structure of tetramer may come from the Y-shaped trimer, to which a fourth particle is inserted. However, evidenced by confocal images, the facet of the fourth particle may not have a full contact with either the substrate or other particles. It should be noted that the formation of tetramer may come from the co-assembly between **S3** and other impurities, like **S2** (Supplementary Fig. 9 and Supplementary Note 2 for more discussion).

The **S4** particles bind to the substrate in a similar manner to **S2** and **S3**. The inequivalent facets are labeled as α, β, and γ (Fig. 6a–e). There are three α and three β facets next to the wetted facet and one γ facet furthest away from the TPM lobe (Fig. 6a, e). Facets β and γ are more accessible to the substrate than facet α, due to steric effects (Fig. 6b). However, orientation with facet γ contacting the substrate is not favored because of the greater gravitational potential (particle standing up) (Fig. 6c). Therefore, only one orientation is preferred, in which the facet β contacts the substrate (Fig. 6d). The confocal microscope image shows result consistent with these hypotheses (Fig. 6f).

The assembly of **S4** also gives rise to chiral dimer, achiral Y-shaped trimer and chiral trefoil trimer (Fig. 6g, Supplementary Fig. 10), where the chiral trefoil trimer dominates (Fig. 6i). Figure 6h highlights the chirality of these assemblies, the origin of which is the same as that of **S3**.

Through synthesis, we can tune the relative size of MOF particle ($\varphi$) and TPM lobe ($d$) for a Janus particle while keeping its configuration. The patch ratio, defined as $r = \varphi/d$, affects the assembly outcome (Supplementary Figs. 11, 12). For example, for **S2**, no assemblies are observed when $r = 0.44$, whereby a big TPM lobe provides strong steric effect preventing the assembly. When $r = 0.81$, chiral clusters are assembled, which is different from the trefoil trimer by particles with $r = 0.64$ (Fig. 4), This suggests that the size of TPM influences the patch contact and thus the chirality of the clusters. For **S3** particles, chiral trimers are observed with a medium size of TPM ($r = 0.74$). When $r = 1.14$, irregular chains are observed due to the small steric effect that cannot prohibit the further extension of assembly.

While a flat substrate is key to the assembly of chiral trimers, we have also explored the assembly of Janus particles in 3D, when a rough substrate is used (Supplementary Fig. 13). For **S1** particles, the assembly gives rise to dimers, whereby the only MOF facet exposed binds one another. For **S2**, an interesting 3D assembly is observed, in which six particles are arranged in a 3D cluster structure (Supplementary Fig. 13, Supplementary Movie 5). The cluster can be interpreted as two **S2**-trefoil trimers attaching to each other (instead of being attached to the substrate). For **S3** particles, clusters of 4–7 particles are formed but they lack uniformity (showing irregular structures). We speculate that, with more exposed facets, the system is kinetically arrested and is difficult to reach the minimum energy state within a reasonable time frame.

### Degree of chirality

There is a noticeable difference in the magnitude of chirality for trefoil trimers assembled by particles of different configurations. Specifically,

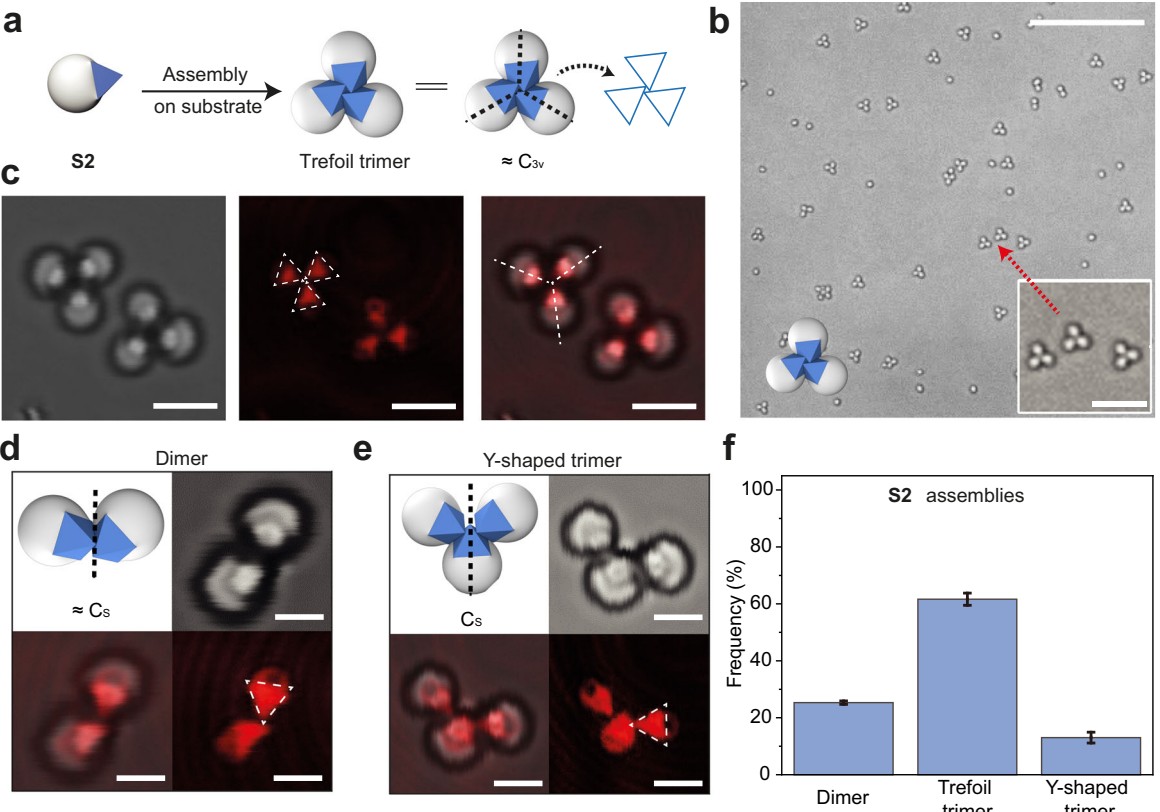

**Fig. 4 | The assembly of S2. a** Cartoon illustration of the assembly of **S2** Janus particles into trefoil clusters with a C$_{3v}$ symmetry. **b** Bright-field microscope image (zoomed-out view) of **S2** assemblies; inset shows the magnified trefoil clusters. **c** Confocal microscope images (transmission, reflected, and overlay) of the trefoil clusters. The orientation of the particles can be deduced by triangular pattern observed, and the cluster symmetry is also highlighted. Confocal microscope images of dimer (**d**) and Y-shaped trimer (**e**) assembled from **S2** particles. **f** The number frequency of dimer, trefoil trimer and Y-shaped trimer among the **S2** assemblies. Error bars are standard deviations. The measurement for frequency is repeated 5 times and at least hundreds of particles are measured each time. Scale bars: 25 μm for (**b**), 5 μm for inset image in (**b**), 3 μm for (**c**), 1.5 μm for (**d, e**).

the trefoil trimer based on **S2** with an intermediate patch ratio ($r = 0.64$) appears to lack obvious chirality, while trefoil trimers by **S3** and **S4** (with comparable $r$) exhibit increased chirality. To facilitate a quantitative analysis, we utilize the Continuous Symmetry Measures[37] to assess the degree of chirality (DC) of the trimers, which we described in 2D as combination of circles and triangles (Fig. 7). The DC is defined as

$$DC = \min\left(\frac{\sum_{i=1}^{N}|\mathbf{q_i} - \mathbf{p_i}|^2}{\sum_{i=1}^{N}|\mathbf{q_i} - \mathbf{q_0}|^2}\right) \times 100 \qquad (1)$$

where $\mathbf{q_i}$ refers to the position vectors of points in the 2D model of the chiral trimers, $\mathbf{p_i}$ refers to the position vector of points in the achiral object as the reference, and $\mathbf{q_0}$ refers to the position vector of the geometric center of the chiral object (Fig. 7a, b, see Methods for more details). A DC value of zero means that the structure is not chiral, whereas a greater value suggests a greater chirality. The DC value calculated for the **S2** trefoil trimer is 0.95 (Fig. 7c), while the values for the **S3**- and **S4**-based trefoil trimers are DC = 57.69 (Fig. 7d) and DC = 53.49 (Fig. 7e), respectively. This indicates that the **S3**- and **S4**-trimers have significantly greater chirality compared to the **S2** trefoil trimer. Notably, the **S3** and **S4** trefoil trimer have similar DC, with their chirality mainly originates from the arrangement of the UiO-66 octahedra. The fact that **S4** trimer is slightly less chiral than that of **S3** may be attributed to the more remote TPM lobes. These differences are eventually due to the variations in the shape of Janus particles based on the extent of dewetting.

## Monte Carlo simulation of assembly

We perform Monte Carlo simulation (using HOOMD-blue) to explore the assembly of Janus particles. The implicit depletant simulation algorithm, previously developed by Glaser et al., is used to simulate depletion interaction between anisotropic colloids in an implicit way[38]. In this algorithm, one can control the depletant size and depletant fugacity to change the magnitude of depletion force between large particles. The fugacity, $f$, is defined as $f = e^{\beta \mu_p} / \lambda_p^3$, where $\mu_p$ denotes the chemical potential of the depletants, $\beta$ denotes the reciprocal of the product of the Boltzmann constant $k_B$ and the temperature $T$, and $\lambda_p$ denotes the de Broglie wavelength. The fugacity is a function with positive correlation to the number density of depletants, whereby a higher value of $f$ is associated with a stronger depletion interaction[38].

To verify the model, we first simulate the selective assembly of octahedral particles in the presence of spheres. The ratio between the radius of the sphere and the size of octahedron ($\varphi$) is set as 0.7, and the ratio between the size of sphere and the size of depletants is 20. All particles are confined to move only on the bottom wall of a 3D box, mimicking the behavior of particles on a substrate in experiments. For octahedral particles, one facet is set to be parallel to the bottom. The particles are allowed to translate only in the $xy$-plane and rotate only around the $z$-axis. We have determined that when $f = 250$ to 400, the octahedral particles can assemble, while the spheres do not, due to weaker interactions (Fig. 8a, Supplementary Fig. 14a).

We then simulate the assembly of Janus particles. The particle models are built by connecting a sphere to an octahedron with or without overlap, roughly replicating actual particle shapes. The particles are confined to the substrate (i.e., with one facet of the octahedron

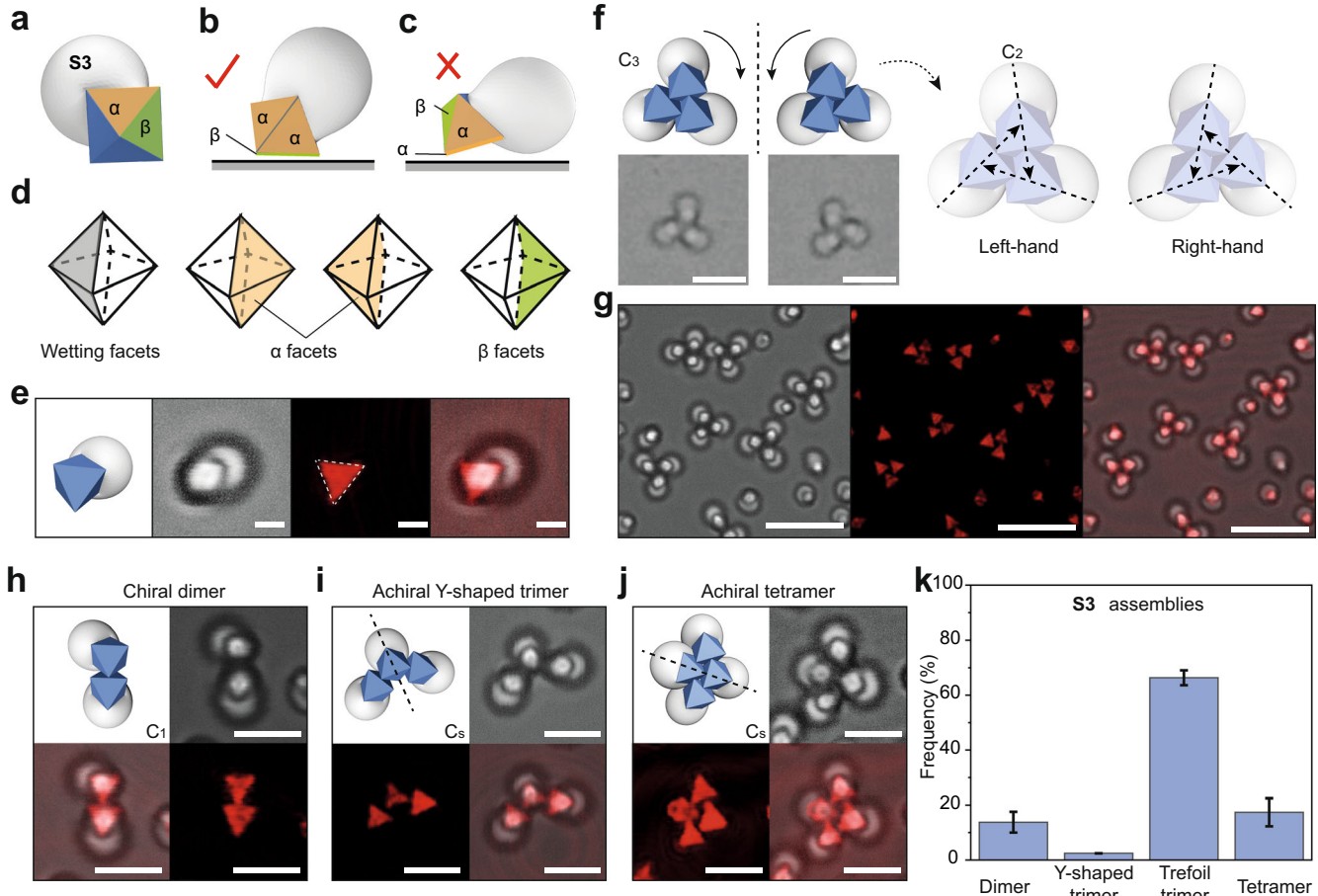

**Fig. 5 | Assemblies of S3 into chiral clusters. a–c** Cartoon showing the two types of facets of **S3** (labeled as α and β). Due to steric hindrance of TPM, **S3** particles prefer to sit on the β facet (**b**, shown in green) rather than α facet (**c**, shown in yellow). **d** Cartoon showing the inequality of MOF facets of **S3**. The dashed lines refer to the back side of UiO-66 and the solid lines refer to the front side. **e** Confocal micro-scope images (transmission, reflected, and overlay) of individual **S3** particle on the substrate. **f** Cartoon and bright-field images of chiral trimers assembled from **S3**. The chirality of **S3** assemblies comes from the arrangement of three $C_2$ axes of symmetry in a clockwise (left-handed) or counterclockwise (right-handed) direc-tion. **g** Confocal microscope images (transmission, reflected, and overlay) of chiral trimers. Chiral dimer (**h**), achiral Y-shaped trimer (**i**), and achiral tetramer (**j**) are shown in confocal microscope images. **k** Statistic number frequency of dimer, achiral trimer, chiral trimer, and tetramer among **S3** assemblies. Error bars are standard deviations. The measurement for frequency is repeated 6 times and at least hundreds of particles are measured each time. Scale bar: 1 μm for (**e**), 5 μm for (**f**), 8 μm for (**g**), 3 μm for (**h**–**j**).

parallel to the substrate). The assembly of **S3** and **S4** is tested first, which successfully produces chiral clusters when $f = 350$ (Fig. 8b, Supplementary Fig. 15a), consistent with the experimental results (Figs. 5, 6). The chiral clusters have both handedness. Various other clusters including the dimer and achiral trimer are also observed (Fig. 8c, Supplementary Fig. 15b).

We next simulate the assembly of **S2**, where the steric effect of the TPM lobe have shown to play an important role. To evaluate this effect, the size of the sphere in the model is varied to realized different patch ratios ($r$) of the Janus particles (Fig. 8d). While we have obtained (chiral) trimer clusters, the DC value decreases, from DC = 13.77 to 3.04, as the patch ratio decreases, from $r = 0.66$ to $0.56$, and no assembly when $r = 0.54$ (Fig. 8e, f, Supplementary Fig. 14b). This result is consistent with the trend observed in our experiments, wherein **S2** trimers exhibit obvious chirality when $r = 0.8$, small chirality when $r = 0.64$, and few assemblies when $r = 0.44$ (Supplementary Fig. 11). The differences in chirality can be appreciated by the size of the triangle in the center of the cluster, due to the different arrangement of the octahedral parti-cles (Fig. 8e). The variation is attributed to the different steric effects exerted by the TPM lobes. Large TPM lobes prevent the MOF patches from maximizing their contact area, resulting in low DC values, while small TPM lobes allow for the increased patch contact and high DC values.

## Discussion

We have introduced the concept and synthesis of a class of Janus particles, whose patches adopt various tunable shapes and symme-tries. Crucial to the strategy is a controllable dewetting process that modulates how an oil droplet contacts and encapsulates a polyhedral MOF particle. Surface energy minimization, revealed by simulation, dictates the overall particle shapes. The employment of polyhedral particles also plays a vital role in the site-selective assembly of the Janus particles through depletion interaction. The assembled structures are a result of the shape characteristics of the Janus particles.

While chirality is an important concept in the molecular world, it has significant implications in colloidal materials. Previously, chiral colloidal structures (individual particles or their assemblies) have exhibited high sensitivity to circular polarization[39] or have been uti-lized for manipulating immunological responses[40]. Different from previous approaches[5,41–43], we realize colloidal chirality by self-assem-bly, which emerges from achiral building blocks due to their shape, patch symmetry, and substrate confinement. The DC can also be tuned. Looking forward, out findings on controlling particle symmetry should inspire the design of colloids and intricate colloidal assemblies. For example, now that we have focused on octahedral particles, other polyhedral shapes may be used to create different patch and particle symmetries.

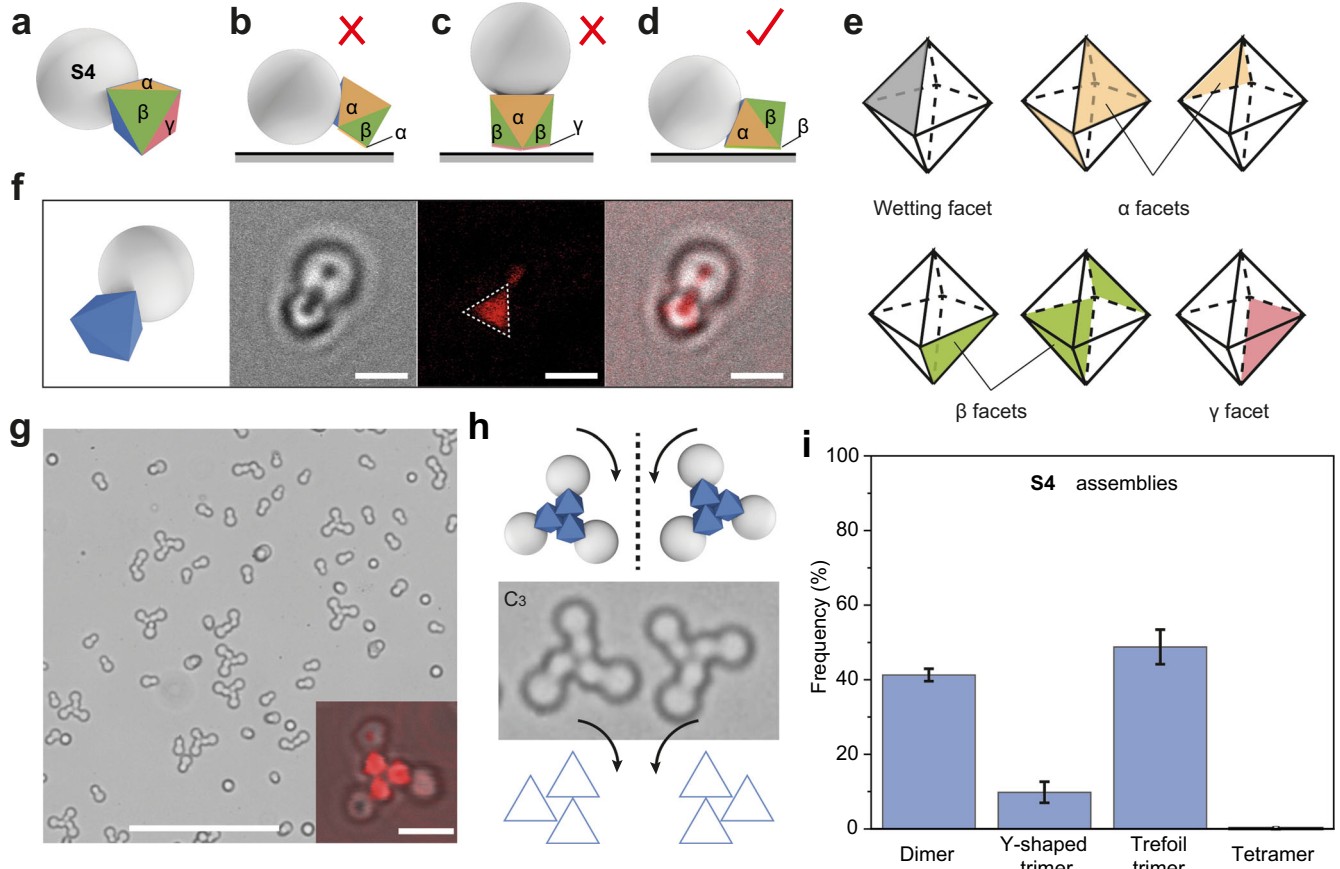

**Fig. 6 | The assembly of S4 Janus particles. a–d** Cartoon showing the types of facets of **S4** (labeled as α, β, and γ). The possible orientations of **S4** with respect to the substrate, where facet α (**b**, shown in yellow), β (**d**, shown in green), or γ (**c**, shown in red) contacts the substrate. Due to steric hindrance of TPM and gravitational potential, **S4** particles prefer to sit on the β facet (**d**) rather than the other two facets (**b**, **c**). **e** Cartoon showing the inequality of MOF facets of **S4**. The dashed lines refer to the back side of UiO-66 and the solid lines refer to the front side. **f** Model and confocal microscope images (transmission, reflected, and overlay) of individual **S4** particles on the substrate. **g** Bright-field optical image of assemblies by **S4** particles. Inset shows the chiral trefoil trimers. **h** Cartoon and optical microscope images of **S4** trefoil clusters, highlighting their chirality. **i** Statistic number frequency of **S4** assemblies of dimer, achiral trimer, chiral trimer, and tetramer. Error bars are standard deviations. The measurement for frequency is repeated 5 times and at least hundreds of particles are measured each time. Scale bars: 2 µm for (**f**), 25 µm for (**g**) and 3 µm for the inset image in (**g**).

## Methods

### Materials
Zirconyl chloride octahydrate (ZrOCl$_2$ · 8H$_2$O) (Dieckmann, 98%); terephthalic acid (H$_2$BDC) (J&K, 99%); dimethylformamide (DMF) (RCI Labscan, 99.9%); acetic acid (RCI Labscan, 99.8%); 3-(trimethoxysilyl) propyl methacrylate (TPM) (Sigma-Aldrich, ≥ 97%); ammonia solution (Scharlau, 25% in water); Synperonic F-108 surfactant (F-108) (Sigma-Aldrich); cetrimonium chloride (CTAC) (Dieckmann, 98%); Darocur 1173, Ciba (Energy Chemical, 98%); Triton X-100 (TX) (Alfa Aesar); 2,2′-azobis(2-methylpropionitrile) (AIBN) (Sigma-Aldrich, 98%).

### Synthesis of UiO-66 microcrystals
Micrometer-sized UiO-66 crystal were synthesized based on previous researches[44]. In brief, 1 g of ZrOCl$_2$ · 8H$_2$O and 2 g of H$_2$BDC were mixed in a 200 mL vial before addition of 60 mL of DMF and 40 mL of acetic acid. The mixture was ultrasonic treated for 30 min before it was heated at 90 °C for 48 h. The obtained UiO-66 particles were washed with DMF and ultrasonic treated several times to remove residues. Then UiO-66 was washed with DI water three times to remove DMF and stored in water.

### Synthesis of Janus UiO-66-TPM particles
500 µL aqueous suspension of UiO-66 particles were mixed with 2 mL of water, 25 µL of TPM and 12.5 µL of ammonia aqueous solution (5.6%). The mixture was stirred at 200 rpm for at least 40 min until UiO-66 particles were encapsulated with oil droplets, observable under an optical microscope. After that, trace amount of 3 mM F-108 aqueous solution was added to stabilize the TPM oil by forming polymer brush on its surface, before the mixture was centrifuged at 500 rpm (or centrifugal force of approximate 20 × $g$) for 10–20 min, making sure the centrifugation was gentle as possible to avoid merging of the droplets. The supernatant was decanted, and the centrifuged pellet was dispersed in pure water. This wash process was repeated several times. Then, a desired amount of 10% v/v Triton X-100 aqueous solution was added to change the particle shape by triggering the dewetting of the TPM oil. Once the desired geometry was formed, trace amount of AIBN was added and the reaction mixture was heated at 80 °C for at least 4 h to initiate radical polymerization. The polymerization process can be achieved alternatively by UV light, when a photo-initiator, Darocur 1173, Ciba was loaded into the reaction (1 wt.%) before TPM coating on UiO-66. Once the desired geometry of Janus UiO-66 was obtained, the sample was exposed to UV (377 nm, 7800 mW cm$^{-2}$) for 30–60 min. The obtained Janus UiO-66 was washed several times by centrifugation/redispersion cycles in water.

### Assembly of Janus UiO-66
The suspension UiO-66 particles and concentrated CTAC aqueous solution were mixed and the final CTAC concentration is 3–6 mM. The

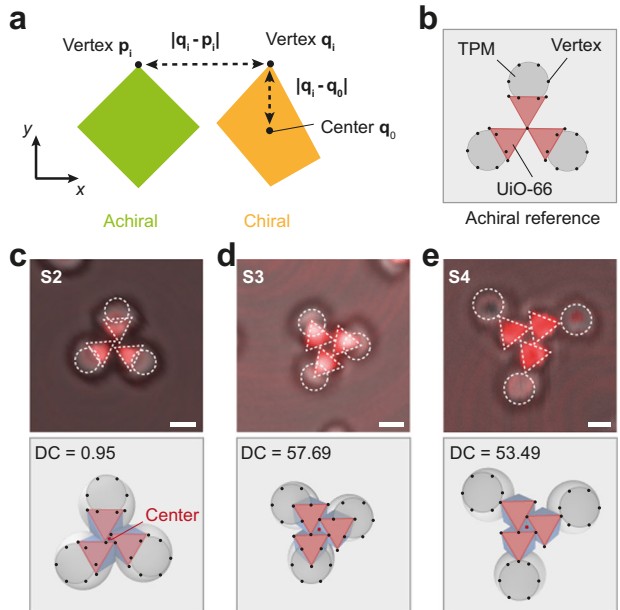

**Fig. 7 | Degree of chirality. a** Illustration of calculation for degree of chirality (DC). The chiral object is presented by yellow, and the achiral reference is represented by green. **b** The achiral reference (hypothetic) of a trefoil trimer. The TPM lobe is represented as a gray cycle and UiO-66 is represented as a light red triangle. The vertex for calculation is presented as a black dot. Six black dots are selected from a TPM sphere, and three black dots are picked from a UiO-66 triangle. Illustration of trefoil trimer of **S2** (**c**), **S3** (**d**) and **S4** (**e**) and the corresponding confocal overlay images. The center is represented as a red dot in the middle of each trimer. The DC values of each assembly are provided. Scale bars: 1 μm for (**c**–**e**).

suspension was then injected into a glass capillary tube for assembly. The assembly process was observed under an optical or confocal microscope. The **S2** and **S3** particles used for confocal characterization are shown in Supplementary Fig. 16 with more details. The size of Janus UiO-66 particles does not influence the assembly results (Supplementary Fig. 17, Supplementary Note 3).

### Surface Evolver simulation
Surface Evolver[32] was used to simulate the geometries of Janus particles in different state. Surface Evolver evolves the liquid surface towards minimal energy level via a gradient descent method of surface tension and works for the liquid wetting and dewetting phenomenon. Under the constraints dictated by the geometry of octahedra (UiO-66), the pre-defined initial liquid surface (TPM) was evolved toward the minimal energy, in which each cell on the surface is susceptive to refinement by a triangular tessellation to achieve the desired level of accuracy. Each iteration step in the Evolver minimized the energy by changing surface in the negative gradient direction of surface energy, until reaching a global minimum.

### Confocal microscope characterization
Confocal laser scanning microscope (Leica TCS SP8) was utilized to observe the orientation of Janus UiO-66 particles and their assembly on the glass substrate. Samples were charged into a glass capillary tube, which was sealed to prevent solvent evaporation and mounted on a glass microscope slide. For the reflected mode microscopy to observe the particle facet facing the substrate, the photomultiplier tube was set to collect the light with the identical wavelength from the incident laser.

### Zeta potential measurement
TPM oil droplets were washed and re-dispersed in de-ionized water (dilute sample, approximately 0.01 mg mL⁻¹). The zeta potential is measured by Zeta Potential Analyzer (Zetasizer, Malvern).

### Interaction potential model
The theoretical interaction potential between colloidal UiO-66 particles and TPM particles is treated as a sum of depletion interaction (dp), van der Waals interaction (vdW) and electrostatic repulsion (el).

$$U_{tot} = U_{vdW} + U_{el} + U_{dp} \tag{2}$$

For simplicity, the facet of UiO-66 particles is approximated as a flat plane and TPM as a sphere. Then, the interaction between UiO-66 and UiO-66, UiO-66 and TPM, TPM and TPM are modeled by plane-to-plane, plane-to-sphere, and sphere-to-sphere systems, respectively.

The van der Waals interactions for each system are given by intermolecular and surface forces[45]:

$$U_{vdW}^{plane-plane} = -\frac{HA}{12\pi d^2} \tag{3}$$

$$U_{vdW}^{plane-sphere} = -\frac{HR}{6d} \tag{4}$$

$$U_{vdW}^{sphere-sphere} = -\frac{H}{6d}\left(\frac{R_1 R_2}{R_1 + R_2}\right) \tag{5}$$

Where $d$ is the separation between particle surfaces, $A$ is the plane-to-plane overlap area, $R$, $R_1$ and $R_2$ are the radius of sphere and taken with identical value for convenience, $H$ is the Hamaker constant calculated by the Hamaker constant of water and MOF according to the following formula[46]:

$$H_{123} \approx \left(\sqrt{H_{11}} - \sqrt{H_{22}}\right)\left(\sqrt{H_{33}} - \sqrt{H_{22}}\right) \tag{6}$$

Where the number subscript refers to the material of interest. The electrostatistic interactions for each system are given by:

$$U_{el}^{plane-plane} = \frac{\kappa}{2\pi} Z e^{-\kappa d} A \tag{7}$$

$$U_{el}^{plane-sphere} = RZ e^{-\kappa d} \tag{8}$$

$$U_{el}^{sphere-sphere} = \left(\frac{R_1 R_2}{R_1 + R_2}\right) Z e^{-\kappa d} \tag{9}$$

Where $\kappa$ is the inverse Debye length given by:

$$\kappa = \left(\frac{ce^2}{\varepsilon\varepsilon_0 k_B T}\right)^{1/2} \tag{10}$$

Where $c$ is the ionic strength, $e$ the elementary charge, $k_B$ Boltzmann's constant, $\varepsilon$ the relative permittivity, $\varepsilon_0$ the permittivity of the free space, and $T$ the temperature. The ionic strength is calculated according to the previous report[47].

The value of $Z$ is defined as:

$$Z = 64\pi\varepsilon\varepsilon_0 \left(\frac{k_B T}{e}\right)^2 \tanh^2\left(\frac{ze\psi_0}{4k_B T}\right) \tag{11}$$

Where $\psi_0$ is the surface potential and $z$ the electrolyte valency. Depletion interactions are given by previous report[42,48]:

$$U_{dp} = -\Delta\Pi \times V_{OV} \tag{12}$$

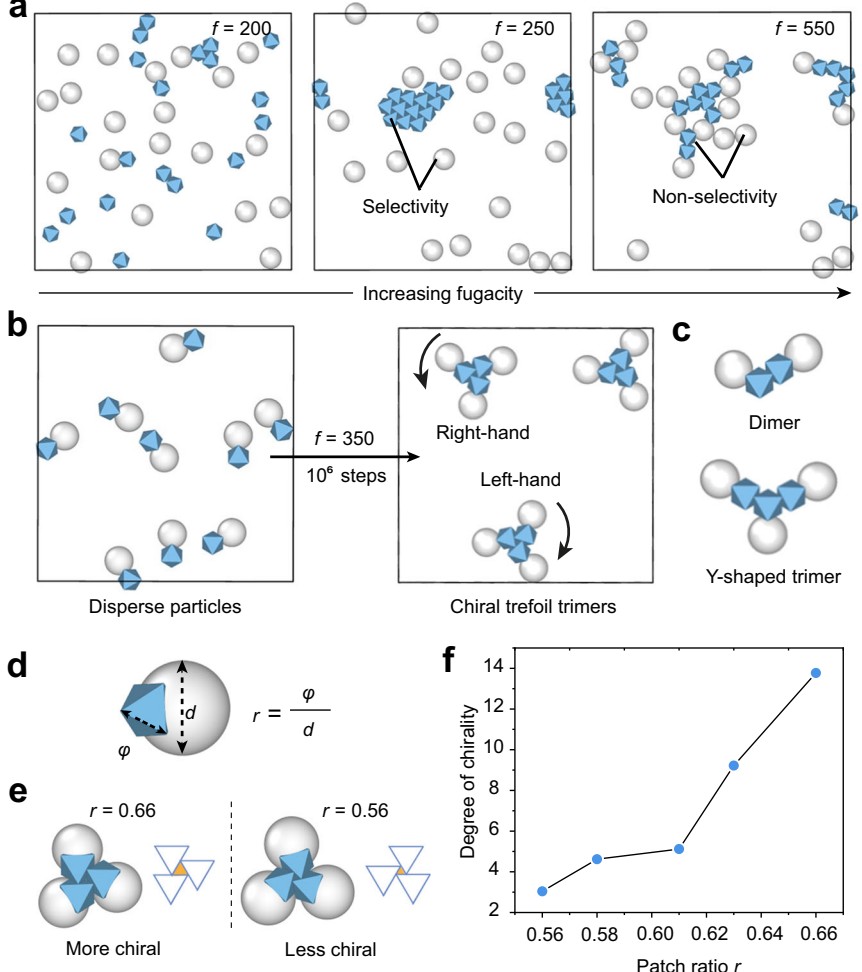

**Fig. 8 | Monte Carlo simulation of self-assembly. a** Selective assembly of UiO-66 particles (shown in blue) and TPM particles (shown in gray) in different settings of fugacity (*f*). **b** Self-assembly of **S4** when *f* = 350 arb. unit. **c** Clusters of **S4** by simulation. **d** Illustration of the definition of *r*. **e** Self-assembly of **S2** with different patch ratios (*r*) when *f* = 350 arb. unit. The differences between the assembled clusters can be revealed by the size difference of the triangle (orange) in the center of the cluster, as a result of the different arrangement of the octahedral particles (blue). **f** The degree of chirality (DC) of **S2** trimer with different values of *r*.

Where $\Delta\Pi$ is the change in osmotic pressure given by:

$$\Delta\Pi = n k_B T \left(1 + \phi_{eff} + \phi_{eff}^2 - \phi_{eff}^3\right)\left(1 - \phi_{eff}\right)^{-3} \quad (13)$$

Where $n$ is the concentration of depletant and defined as $n = (N_A/N_{agg})(C_{CTAC} - CMC)$, $N_A$ is Avogadro's number, $N_{agg}$ is the aggregation number for CTAC micelle, $C_{CTAC}$ is the concentration of CTAC and CMC is the critical micelle concentration of CTAC (1.31 mM). $\phi_{eff}$ is defined as $\phi_{eff} = n(4/3)\pi(D_{eff}/2)^3$.

$V_{OV}$ is the overlapping volume and give by colloids and the depletion interaction[49]:

$$V_{OV}^{plane-plane} = A\left(2t_{eff} + D_{eff} - d\right) \quad (14)$$

$$V_{OV}^{plane-sphere} = \frac{\pi}{3}\left(D_{eff} + 2t_{eff} - d\right)^2\left(3R + \frac{D_{eff}}{2} - 2t_{eff} + d\right) \quad (15)$$

$$V_{OV}^{sphere-sphere} = \frac{\pi}{6}\left(D_{eff} + 2t_{eff} - d\right)^2\left(3R + D_{eff} - t_{eff} + \frac{d}{2}\right) \quad (16)$$

Where $D_{eff}$ is the effective size of the CTAC micelle diameter and $t_{eff}$ the effective size of the CTAC bilayer thickness on particle surface, both of which are calculated based on the previous methods[50].

**The degree of chiralit**

The DC[37] of trefoil trimers is defined by Eq. (1).

Due to the complex shape of Janus UiO-66 particle, we simplify the trimer into a 2D geometric model, with triangles denoting the UiO-66 patches and circles denoting the TPM lobes, for convenient analysis. As shown in Fig. 7b, we first set an achiral trimer as a reference, which serves as $p_i$, then we displace the coordinates of triangles and spheres to mimic the trimers of **S2, S3** and **S4**, which serve as $q_i$, as shown in Fig. 7c–e. There are 9 vertexes picked up from a single Janus particle, where three vertexes from the triangle (UiO-66 patch) and 6 vertexes from the sphere (TPM lobe). The center $q_0$ is set in the middle of each trimer.

We admit it that this model may not be very accurate because of the geometric simplifications. For example, TPM lobe should be dome-shaped instead of sphere, and the trimer is not a 2D model but 3D. However, such simplification is justified as it at least gives us a quantitative measurement of the chirality degree of each trimer and thus demonstrate the tunability of Janus UiO-66 particle in controlling the assembly structure.

**Monte Carlo simulation**

Monte Carlo Simulations were performed using the hard particle Monte Carlo method (HPMC) implemented in the modified HOOMD-blue simulation package[51,52]. Two types of systems with implicit

depletants[38] were constructed to investigate the selective interactions and chiral clusters respectively.

In the selective interaction simulations, the simulation box of 15 $\sigma \times 15\ \sigma \times 15\ \sigma$ contained 20 spherical particles and 20 octahedron particles. The radius of sphere was 0.855 $\sigma$ and the side length of the octahedron was 1.22 $\sigma$. All particles were positioned close to the bottom wall of the simulation box (for octahedron particles, one face was close to the bottom). To explore the selective interaction, we set the size ratio between sphere particles and depletants to be 20:1 (0.855:0.04275), with different fugacity values.

In the Janus particle simulations, each particle consisted of a sphere and an octahedron, with one face (bottom) of the octahedron close to the bottom plane of the simulation box. According to the particle shapes in the experiment, the sphere was positioned tangent to the octahedron's side plane (S4) or overlap with octahedron partially (S2, S3). In S3 and S4 model, the sizes of sphere, octahedron and box were the same as mentioned above, with the size of sphere that can be changed in S2. The main size ratio between sphere particles and depletants used in all system was 20:1. Each system with various depletant fugacities contained three Janus particles to observe the formation of cluster. Based on the preliminary results, simulations were also performed for systems containing 9 particles with the depletant radius of 0.04275 $\sigma$ and depletant fugacity of 350 $\sigma^3$ to investigate the chirality of clusters.

In the HPMC simulations of modified HOOMD package, particles were confined to move only on the bottom wall of the box, mimicking the behavior of particles on a substrate in experiments. This constraint allowed particles to translate only in the $xy$-plane and rotate only around the $z$-axis. All simulation results were obtained within $1-2 \times 10^6$ Monte Carlo steps in the NVT ensemble at the unit temperature.

## Reporting summary

Further information on research design is available in the Nature Portfolio Reporting Summary linked to this article.

## Data availability

The data that support the findings of this study are available from the corresponding author upon request.

## Code availability

The codes used for the simulation are available from the corresponding author upon request.

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

## Acknowledgements

We thank Prof. Ken Brakke (Susquehanna University, USA) for his kind help on Surface Evolver simulation and for reading and editing the manuscript. This project is partially supported by the General Research Fund (No. 17314222, to Y.W.) and Collaborative Research Fund (No. C7075-21G, to Y.W.) from Research Grant Council (RGC) of Hong Kong, Croucher Innovation Award 2019 from the Croucher Foundation of Hong Kong (to Y.W.), and NSFC Excellent Young Scientists Fund 2020 (Hong Kong and Macau, No. 2022206, to Y.W.). T.Z. and D.L. acknowledge the support from university postgraduate fellowship. X.F. and R.N. acknowledge the support from the Academic Research Fund from Singapore Ministry of Education Tier 1 Gant (RG59/21). Publication was made possible in part by support from the HKU Libraries Open Access Author Fund sponsored by the HKU Libraries.

## Author contributions

Y.W. conceived and supervised the project. T.Z. and W.X. synthesized the Janus UiO-66 particles. T.Z. and D.L. performed the self-assembly using depletion interaction. T.Z. and D.L. performed confocal imaging. T.Z. simulated the geometries of Janus UiO-66 particles using Surface Evolver. T.Z., D.L and W.X. participated in discussion and analysis of the experimental data. X.F. and R.N. performed the HOOMD-blue simulation. T.Z. and Y.W. wrote the manuscript with revision from all authors.

## Competing interests

The authors declare no competing interests.
