## [Peer Review File · Nature Communications]

Janus Particles with Tunable Patch Symmetry and their Assembly into Chiral Colloidal ClustersREVIEWER COMMENTS

Reviewer #1 (Remarks to the Author):

This paper describes a new kind of Janus colloidal particle consisting of an octahedral MOF particle combined with a polymerizable liquid oil (TPM) droplet. The authors identify four different configurations of combined particles in which a different number of facets of the octahedral particle are exposed. The number of exposed facets is controlled by the concentration of surfactant used when the oil droplet and octahedral particle are combined. The four different Janus configurations are labeled S1, S2, S3, and S4.

It would be helpful if the authors would indicate how monodisperse the MOF particles and oil droplets were. Both here and in the later sections on self-assembly, I wonder how much the observations depend on the relative sizes of the MOF particles and the oil droplets, particularly for the self-assembly where steric hindrance due to the size of the spherical lobe plays an important role in what kinds of structures form. It would also be interesting to know the degree to which the absolute size of the particles can be varied as this sets how the particles interact with light and other fields.

The paper identifies a spectrum of 2D self-assembled structures. In all cases, the depletion attraction between the faceted MOF face and the substrate plays a dominant role. Perhaps the most interesting structures are the chiral assemblies that form. Both left- and right-handed assemblies form, presumably in equal numbers to within statistical uncertainty. The authors should take care, then, in claiming regioselective assembly (pages 2, 18, and 21). As far as I can ascertain, there is nothing that preferentially selects right or left-handed assemblies. That is, the process is not regioselective. Indeed, one would be shocked if it were as the particles themselves are not chiral, as the authors take pains to emphasize.

I was disappointed that the paper did not explore self-assembly between particles without the influence of depletion attraction to the substrate. This should be possible simply by roughening the substrate, which can be done in any number of ways. This would be interesting as it would point the way to what the possibilities might be for 3D self-assembly of these particles.

Finally, I am left wondering what the motivation is for making any of these structures, chiral or otherwise. Some vague reference to photonics, catalysis, and liquid transport is made in the opening sentence, but I would hope for a clearer vision of some set of target structures that have a clear value in some relevant context.

Reviewer #2 (Remarks to the Author):

This manuscript reports on the synthesis and assembly of hybrid particles. The latter ones are made by nucleation and polymerization of monomer droplets onto hexagonally-shaped MOF microparticles. Wetting and related to this immersion of the octahedral seeds into the oil can be nicely controlled by the concentration of added surfactant. This enables the synthesis of dimer-like microparticles with various defined symmetries. Most remarkably, the complex particles exhibiting a faceted half can be assembled into chiral particle clusters. Synthesis of the hybrid particles and the study of the shape-mediated superstructure formation mark the definitive strength and innovation of this manuscript. However, the manuscript has several weaknesses that should be removed to further strengthen the relevance of the manuscript.

i) The structure of the manuscript requires improvements. It is rather uncommon and somewhat misleading that the method section is embedded between two different bibliographies. I initially believed that the method section and the second bibliography had been provided as supplementary

information before I saw the true supporting information.

ii) For the sake of clarity, the title should better be rewritten as follows: "Janus Particles with Tunable Patch Symmetries and their Assembly into Chiral Colloidal Cluster". The present title indicates that two independent subjects are studied.

iii) The terms "dome-shaped" should be defined. Furthermore, the term patch describes a limited area on the surface with specific properties whereas the present study widely reports on hybrid particles that exhibit a faceted half.

iv) The synthesis of the UiO-66 particles used in this study was already reported by other authors. A reference to these reports is missing.

v) Line 91-92: Nucleation of TPM droplets on the MOF particles is explained by oppositely charged zeta potentials. How was the value of -30 mV for the TPM droplet determined? The manuscript does not contain any information on zeta potential measurements. How was the zeta potential calculated? Why should a monomer nucleus already show a zeta potential that can explain the nucleation?

vi) Figure 6: A more detailed discussion on the inequality of the MOF facets would be helpful to better understand the mechanism behind superstructure formation.

vii) Why do the TPM droplets attached to the MOF particles survive the rather harsh centrifugation, pelleting and re-dispersion treatment?

vii) Surface evolver simulation was certainly not established by these authors. References to earlier reports on the techniques are missing.

viii) Modelling of the degree of chirality; a schematic representation that shows the chosen geometric model would be helpful to readers.

ix) The manuscript contains several typos that should be corrected.

Reviewer #3 (Remarks to the Author):

The Nature Communications article by Zhang details the synthesis and self-assembly of Janus particles composed of polyhedral MOF and polymeric TPM using a method published by this group in 2022. This method allows for the precise control of the shape utilizing wetting/de-wetting stimulated via Triton X-100. The particles presented exhibit reduced symmetries due in part to the polyhedral MOF used, which is unique to the field and possess the potential to use other similar MOF as a means to access other symmetries. The article explains the presence of only four distinct products after dewetting and thoughtfully used simulations to rationalize these distinct products. Thru the use of simulations, they have elucidated the step-wise asymmetric de-wetting which is entirely new since most Janus particle in literature experience a symmetric de-wetting process. This highlights the possibility that other polyhedron may utilize this same step-wise asymmetric de-wetting. The authors use a combination of shape and depletion forces to achieve the self-assemblies, this is a common practice and is well documented in the literature. The assemblies of Janus particles S2-4 result most notably in chiral clusters, along with others. The authors further defined the degree of chirality of the resultant assemblies. And presented a thorough and thoughtful discussion of the resultant assemblies. Overall, the manuscript is well-written and easy to follow. The manuscript puts the created Janus particles presented into context with current state of the field, as well as builds a solid case that this method could work for other polyhedrons beyond the UiO-66 octahedral used here. Additionally, the authors presented a thorough and thoughtful discussion. The science is very interesting. Congrats to the authors. Therefore, after the minor corrections highlighted below I recommend publication of this

article.

Comments:

1. Duplicate word in sentence (line 36-38) of surprisingly.
2. The abstract would benefit from saying what type of polyhedral MOF is used (UiO-66), as well as specifying the polyhedral as an octahedral. Because this paper only discusses one polyhedral specifically the octahedral UiO-66 and has not proven any other polyhedral shapes to use the same step-wise pathway presented later.
3. A few sentences as an explanation on the difference in affinity between the TPM and UiO-66 to Triton X-100 would be beneficial.
4. Figure 2a would benefit from sub-headers i (top), ii (middle) and iii (bottom) for clarity in the context of the discussion.
5. Lines 155-157, Say that symmetric de-wetting is associated with a higher surface energy but the graph presented in Fig 2c shows the symmetric de-wetting having a lower energy? Is the key in Fig 2c reversed?
6. What informed the choice of depletant? size? charge? What happens if a different depletant is used? Do the assemblies become stereoselective?
7. Fig 3d, the inset is not necessarily adding to the discussion occurring in the rest of Fig 3, as evidenced by the lack of discussion in the text. Could be removed or moved to serve more of a purpose in later figures.
8. The discussion of the antiparallely needs to be more flushed out or discarded. The associated Figure 5f is not adding value to the figure and could be moved to the SI.
9. Fig 5g is not necessary to show the hexagonal superlattice because it's not adding to the discussion of the findings since it's an impossible lattice to achieve with the designed particles as noted Line 283-285.
10. Fig 5h-j lacks the symmetry planes (dashed lines) presented in previous figures, adding them in would be beneficial reader clarity during the discussion.
11. Line 283-285, states that you can only have 3 particles in a cluster but Fig 5j presents a tetramer cluster? So, can you only have max 3 in an assembly or do the TPM portions cause too much steric hinderance to form extended structures? These sentences just need some rewording so the authors aren't contradicting themselves.
12. Figure 6 is the only assembly figure to not present a number percentage of each species that forms, please add.

Reviewer #4 (Remarks to the Author):

In this study, Wang et al. present a novel category of particles called Janus particles, which exhibit controllable patch and symmetries through synthetic means. The method involves incorporating a polyhedral MOF particle within a polymerizable oil matrix. The extent of encapsulation is adjusted by controlling the concentration of surfactants and the dewetting process, which determines final shape of the Janus particle. The authors induced self-assembly of these janus particles by employing depletion forces. Intriguingly, they also observed the formation of chiral colloid assemblies using non-chiral building blocks. This study effectively combines the realms of patchy particles and the growing utilization of MOF particles for self-assembly. The authors demonstrate remarkable control over the fabrication of Janus particles and the formation of diverse colloidal clusters, which depend on the symmetry of the Janus particle. I find this study highly compelling, well explained and believe it is well-suited for publication in Nature Communications. However, I do have a few comments for the authors:

1. The authors should introduce and illustrate with examples the concepts of colloidal MOFs and self-assembly of colloidal MOFs in the introduction. Both concepts, basic for this study, are barely touched in the introduction. The authors should see some recent reviews published in Chem. Soc. Rev.

2. It would be beneficial to include more representative SEM images of the different Janus particles on a larger scale.
3. It is unclear why the authors did not present any SEM images of the assembled clusters and whether they are stable only in wet conditions. Clarification on this point would be helpful.
4. Have the authors explored Monte Carlo simulations to support their calculations on the various cluster assemblies? The incorporation of this type of simulations could strengthen their findings.

Point-by-point response to reviewers' comments.

(Reviewer comments are shown in blue.)

Reviewer 1:

This paper describes a new kind of Janus colloidal particle consisting of an octahedral MOF particle combined with a polymerizable liquid oil (TPM) droplet. The authors identify four different configurations of combined particles in which a different number of facets of the octahedral particle are exposed. The number of exposed facets is controlled by the concentration of surfactant used when the oil droplet and octahedral particle are combined. The four different Janus configurations are labeled S1, S2, S3, and S4.

Authors:

We thank the reviewer for evaluating our work and for the constructive comments. In the revision, we have addressed the concerns raised, believing the work is much improved (see details below and in the revised manuscript).

Reviewer 1:

1. It would be helpful if the authors would indicate how monodisperse the MOF particles and oil droplets were. Both here and in the later sections on self-assembly, I wonder how much the observations depend on the relative sizes of the MOF particles and the oil droplets, particularly for the self-assembly where steric hindrance due to the size of the spherical lobe plays an important role in what kinds of structures form. It would also be interesting to know the degree to which the absolute size of the particles can be varied as this sets how the particles interact with light and other fields.

Authors:

We thank the reviewer for the insightful remarks. As suggested, we have included detailed characterization on the size and size distribution the particles, including both the MOF particles and the oil droplets, as well as their influence on the particle self-assembly. We believe the reviewer's concern has been properly addressed.

Size Dispersity of particles:

Generally, the UiO-66 MOF particles and the TPM particles have a narrow size distribution (or are considered monodisperse). Both types of particles have previously been utilized (by us and others) to assemble colloidal crystals (*Nat. Commun.* **13**, 3980 (2022); *Nat. Chem.* **10**, 78-84 (2017); *J. Am. Chem. Soc.* **137**, 10760-10766 (2015); *Nat. Commun.* **6**, 7253 (2015)).

In our case, we used UiO-66 particles 1180 nm in size (measuring the edge length ϕ of the octahedral particle). The size dispersity $D = 14.4\%$. In our revision, we have also synthesized particles of two more sizes, $\phi = 690$ nm, and $\phi = 575$ nm, with $D = 11.8\%$ and 9.1% , respectively (Fig. R1).

Figure R1. The dispersity of UiO-66 particles. **a**, Scanning Electron Microscope (SEM) image of a UiO-66 particle. **b-d**, The size distribution of UiO-66 particles. Scale bar: 500nm for (a).

After encapsulation by TPM oil (i.e., to make the Janus particle) and the subsequent wash steps, the D drops to $D = 8.4\%$, 8.3% , and 5.1% (based on ϕ). Washing by repetitive centrifugation removed particles of small sizes, thereby improving D . The TPM lobes of the Janus particles always have a low dispersity. Taking **S2** for example, D (by diameter of TPM d) = 6.4% , 6.5% , and 4.0% for different MOF particles.

Because particles with $D < 10\%$ could be regarded as monodisperse (*Biomicrofluidics* **10**, 054107 (2016)). Our particles are sufficiently uniform. Fig. R2 shows this information, which is also included in the revised Supplementary Information (SI). The reviewer can also refer to Fig. R10 for a large-view SEM images of Janus particles.

Figure R2. Dispersity based on MOF patches and TPM lobes of Janus particle **S2**. **a-c**, Size distribution of UiO-66 patches. **d-f**, Size distribution of TPM lobes.

Absolute size of the particles:

As mentioned above, apart from the $\varphi = 1180$ nm particles, which we originally used, we have explored the synthesis and assembly of Janus particles based on of UiO-66 particles of smaller sizes, $\varphi = 690$ and 575 nm. Their sizes are closer to the range (i.e., several 100s of nm) for possible optical/light applications in the future. In all cases, Janus particles can be synthesized, including the desired **S2** and **S3** configurations (shown in Figure R3). More importantly, they can assemble to form clusters (including chiral clusters) identical to that of larger Janus particles.

We also note a few points:

- (1) With a smaller particle size, the particle overlap volume is reduced, so the concentration of depletants needed is higher for depletion-based assembly.
- (2) For small particles, i.e., those based on $\varphi = 575$ UiO-66 particles, the purity of **S4** is compromised by possible detachment of the TPM droplet from the MOF particles.
- (3) Because of TPM oil is formed through emulsion nucleation, their sizes have not been reported to be smaller than 100 nm (too small to be stable). The method is not yet suitable for making nanometer-sized Janus particles.

Figure R3. Janus particles based on MOF (UiO-66) particles of different sizes. a, b, Cartoon and SEM images of Janus particles **S2** (a) and **S3** (b) employing UiO-66 particles with different sizes. **c, d,** Optical microscope images of assemblies of **S2** (c) and **S3** (d) synthesized from 575 nm UiO-66. Scale bars: 2 μm for large images and 500 nm for inset images (a, b); 12.5 μm for (c, d).

Relative size of the MOF patch and the TPM lobe:

For convenience, in each state (**S1-S4**), we define the patch ratio (r) of Janus particle as the edge length of UiO-66 particle (φ) over the diameter of TPM lobe (d), that is, $r = \varphi/d$. The value of r describes the relative size between the MOF patch and the TPM lobe.

In our revision, we have synthesized **S2** and **S3** Janus particles with patch ratios $r = 0.4$ to 1.2. Indeed, the r value has a big impact on the assembled structures. Figure R4 and R5 show the results.

For **S2** particles, when $r = 0.44$, that is, a small patch and big TPM lobe, no assemblies are observed. The TPM lobe exerts a strong steric effect that prohibits the assembly of Janus particles. When $r = 0.64$, the particle has a medium patch and medium TPM lobe; assemblies of clusters ($\sim C_{3v}$ trimers) can be observed. The TPM lobe not only prevents further growth of the clusters but also influences the contact of the MOF patch. When $r = 0.81$, with a large patch and a small TPM lobe, chiral trimers are observed. The TPM lobe exerts relatively small steric effect that allows the maximum contact between MOF patches. The trimer may further grow to

form tetramer since the steric effect of TPM lobe is insufficient to prevent further growth of the clusters (Fig. R4).

Figure R4. The influence of patch ratio on the assembly of Janus particles (S2). **a**, The illustration of patch ratio for Janus particles, where ϕ denotes the edge length of UiO-66 and d denotes the maximum diameter of the TPM lobe. **b**, SEM images of S2 with different patch ratios. **c**, **e**, **g**, Optical microscope images showing the influence of the patch ratio on the assembly of S2. **d**, **f**, Cartoon, optical microscope, confocal and overlay images of trimers found in (c) and (e), respectively. **h**, Degree of chirality (DC) of trimers in (d, f). All the assemblies are conducted within the range of 3-6 mM CTAC. Scale bars: 1 μm for (a), 2 μm for large images in (b), 500 nm for inset images in (b), 1.5 μm for (d, f), 25 μm for (c, e, g).

For S3 particles, when $r = 0.50$, no assemblies are observed. When $r = 0.74$, assemblies (chiral trimers) are observed, where the steric effect allows for assembly of trimers but prohibits their further growth. When $r = 1.14$, irregular chains are observed. The difference between the S2- and S3-assembly (chiral trimer for S2 and irregular chains for S3) at large patch ratio is due to the difference in shape (number of facets exposed).

The above information reveals how the patch ratio and the configuration of Janus particles can affect the assembly, which is now included in SI and in the revised manuscript. We note that, for the current paper, the Janus particles $r = 0.6 - 0.7$ are mostly employed for obtaining the desired assemblies. It is also worthy to note that the patch ratio can be used to alter the degree of chirality of S2 trimer, which is further supported in Monte Carlo simulation (Fig. R11).

Figure R5. The influence of patch ratio on the assembly of Janus particles (S3). **a**, The illustration patch ratio of S3, where ϕ denotes the edge length of UiO-66 and d denotes the maximum diameter of the TPM lobe. **b**, SEM images of S3 with different patch ratios. **c**, **e**, **f**, Optical microscope images showing the influence of the patch ratio on the assembly. **d**, Cartoon, optical microscope, confocal and overlay images of irregular chains found in (c). All the assemblies are conducted within the range of 3-6 mM CTAC. Scale bars: 1 μm for (a), 2 μm for large images in (b), 500 nm for inset images in (b), 12.5 μm for (c, e, f), 5 μm for (d).

Reviewer 1:

2. The paper identifies a spectrum of 2D self-assembled structures. In all cases, the depletion attraction between the faceted MOF face and the substrate plays a dominant role. Perhaps the most interesting structures are the chiral assemblies that form. Both left- and right-handed assemblies form, presumably in equal numbers to within statistical uncertainty. The authors should take care, then, in claiming regioselective assembly (pages 2, 18, and 21). As far as I can ascertain, there is nothing that preferentially selects right or left-handed assemblies. That is, the process is not regioselective. Indeed, one would be shocked if it were as the particles themselves are not chiral, as the authors take pains to emphasize.

Authors:

We thank the reviewer for the comment. The reviewer is correct that both left- and right-hand assemblies are formed in equal numbers. We clarify that the term “regioselectivity” was used to refer to the fact that, for the Janus particles, only the MOF facets are capable of bonding via depletion force. This regio- or site-selectivity is essential for the assembly of chiral clusters but does not contain information that guide the preferential formation of an enantiomers. The term “regioselective” has been previously used for similar purposes (*Chem. Soc. Rev.* **52**, 5684-5705 (2023); *Nat. Mater.* **19**, 1354-1361 (2020)). We did not claim any selectivity over the left- or right-hand assembly.

In revising the manuscript, we have used “site-selectivity” instead and state clearly that the left- and right-hand assemblies are equal in numbers. We hope this can erase any confusion.

Reviewer 1:

3. I was disappointed that the paper did not explore self-assembly between particles without the influence of depletion attraction to the substrate. This should be possible simply by roughening the substrate, which can be done in any number of ways. This would be interesting as it would point the way to what the possibilities might be for 3D self-assembly of these particles.

Authors:

We thank the reviewer for this valuable suggestion. In our revision, we have explored assembly of Janus particles in 3D, using a treated rough substrate. We indeed observe interesting assembly results, which we show below and in Fig. R6.

For **S1** particles, their assembly in 3D gives rise to dimers; the only MOF facet exposed binds one another. For **S2**, an interesting 3D assembly is observed, in which six particles are arranged in a 3D cluster structure. The cluster can be interpreted as two **S2** trefoil trimer attaching to each other (instead of being attached to the substrate). The structure emerges presumably due to the maximization of patch facets that stabilize the overall structure. We have also monitored the assembly process, which undergoes a series of transitions from single particle to dimer, trimer, tetramer, irregular cluster and eventually assembly. The process takes at least 20 minutes to several hours for the structure rearrangement. The lengthy formation process may be ascribed to the many intermediate states as particles rearrange to explore the minimum energy state.

For **S3**, clusters of 4-7 particles are observed. However, they lack uniformity (showing different irregular structures). We speculate that the larger number of intermediate states for **S3** assemblies makes it very difficult to reach the minimum energy state within a reasonable time frame, forming those kinetically arrested products.

These results are also added in the revised manuscript.

Figure R6. Assemblies of Janus particles in 3D. **a**, Illustration of the rough substrate and the 3D assembly of Janus particles. **b-d**, Optical microscope and cartoon images of 3D assemblies of S1 (**b**), S2 (**c**) and S3 (**d**). **e**, Optic microscope and cartoon images showing the dynamics of the formation of 3D assemblies of S2. Scale bar: 12.5 μm , and 2 μm for cropped images in (**b-d**), 6 μm for (**e**).

Reviewer 1:

4. Finally, I am left wondering what the motivation is for making any of these structures, chiral or otherwise. Some vague reference to photonics, catalysis, and liquid transport is made in the opening sentence, but I would hope for a clearer vision of some set of target structures that have a clear value in some relevant context.

Authors:

We thank the reviewer for the suggestion, we have revised the opening sentences; we have also added discussion on the potential utility of chiral colloidal structures.

Specifically, the presence of chirality in colloids and colloidal assemblies imparts unique functional properties to the resulting structures. One notable example is the optoelectronic membrane constructed using chiral gold nanoparticles, which exhibits high sensitivity to circular polarization (*Nat. Nanotechnol.* **17**, 408-416 (2022)). Another example is the utilization of chiral nanoparticles for manipulating immunological responses. (*Nature* **601**, 366-373 (2022)).

Understanding the relationship between structure and properties is crucial for the design of novel materials and devices with desired functionalities. However, the current strategies for designing chiral and other useful colloids and colloidal assemblies remain limited. Tuning the interaction between colloidal building blocks is a fundamental solution for controlling the colloidal structures. One approach involves controlling the binding symmetry of colloidal particles, which can influence their orientation and arrangement during assembly. However, in the previous version of Janus particles, the binding symmetry is always infinity (C_∞) and cannot be modulated. In our work, through particle design and synthesis, we have introduced tunable symmetry in colloidal bonding and realized chirality by assembly.

Reviewer 2:

This manuscript reports on the synthesis and assembly of hybrid particles. The latter ones are made by nucleation and polymerization of monomer droplets onto hexagonally-shaped MOF microparticles. Wetting and related to this immersion of the octahedral seeds into the oil can be nicely controlled by the concentration of added surfactant. This enables the synthesis of dimer-like microparticles with various defined symmetries. Most remarkably, the complex particles exhibiting a faceted half can be assembled into chiral particle clusters. Synthesis of the hybrid particles and the study of the shape-mediated superstructure formation mark the definitive strength and innovation of this manuscript. However, the manuscript has several weaknesses that should be removed to further strengthen the relevance of the manuscript.

Authors:

We thank the reviewer for evaluating our work. We are delighted that the reviewer agrees upon the “definitive strength and innovation” of our study. In our revision, we have addressed the weaknesses mentioned. We are grateful that the reviewer has raised those for improving the manuscript.

Reviewer 2:

i) The structure of the manuscript requires improvements. It is rather uncommon and somewhat misleading that the method section is embedded between two different bibliographies. I initially believed that the method section and the second bibliography had been provided as supplementary information before I saw the true supporting information.

Authors:

We thank the reviewer for the comment. We have revised the paper structure according to the Journal style.

Reviewer 2:

ii) For the sake of clarity, the title should better be rewritten as follows: “Janus Particles with Tunable Patch Symmetries and their Assembly into Chiral Colloidal Cluster”. The present title indicates that two independent subjects are studied.

Authors:

Thanks for the suggestion. We have revised it accordingly.

Reviewer 2:

iii) The terms “dome-shaped” should be defined. Furthermore, the term patch describes a limited area on the surface with specific properties whereas the present study widely reports on hybrid particles that exhibit a faceted half.

Authors:

The term “dome-shape” refers to the shape of hemisphere or a spherical cap. Most reported Janus particles have a patch of such shape, i.e., hemispherical and alike, derived from the spherical particles used for making these Janus particles. The patch on our particles is different, being faceted and shaped, derived from polyhedral particles.

We agree with the reviewer that, at least for the **S3** and **S4** particles, they are hybrid dimer particles. However, the particles evolve from **S1**, which is more appropriate to be called a particle with a triangular patch. So “hybrid particle with faceted half” would not work either. Using “patch” allows us to make our point to contrast to spherical Janus particles widely reported in literature.

In revision, while continuing to use “patch”, we have added extra phrases/sentences to describe the shapes (the hybrid dumbbell characteristics), especially for **S3** and **S4**.

Reviewer 2:

iv) The synthesis of the UiO-66 particles used in this study was already reported by other authors. A reference to these reports is missing.

Authors:

Thanks for the suggestion. We have added references regarding the synthesis of UiO-66 particles in the revised manuscript. (*ACS Appl. Mater. Interfaces* **9**, 33413-33418 (2017))

Reviewer 2:

v) Line 91-92: Nucleation of TPM droplets on the MOF particles is explained by oppositely charged zeta potentials. How was the value of -30 mV for the TPM droplet determined? The manuscript does not contain any information on zeta potential measurements. How was the zeta potential calculated? Why should a monomer nucleus already show a zeta potential that can explain the nucleation?

Authors:

The zeta potential of TPM droplets is directly measured by a Zeta Potential Analyzer (Zetasizer, Malvern), which reveals the charge situation on colloid surface.

The monomer TPM contains Si-OMe group that undergoes hydrolysis in aqueous solution, becoming Si-OH. Nucleated by a condensation reaction, the TPM monomer form oil droplets in water, with Si-O⁻ group on the surface electrostatically stabilizing the droplets. The negative surface charge can be experimentally determined. We measured the zeta potential of TPM oil droplets, which range from -20 to -30 mV, in agreement with that of previous research (*Nature* **550**, 234–238 (2017)). We note that the TPM molecules in the droplet still have an arylate group that can further polymerize to solidify the particle.

Reviewer 2:

vi) Figure 6: A more detailed discussion on the inequality of the MOF facets would be helpful to better understand the mechanism behind superstructure formation.

Authors:

Thanks for your suggestion. We have included cartoon to better illustrate the different facets; discussion have also been added. The Figure R7 shows the information:

Figure R7. Illustration of inequality of the MOF facets of **S3** and **S4**.

We thank the reviewer for this suggestion. We have included more cartoon illustration to help demonstrate the different facets of the Janus particles.

Specifically, the **S3** facet (wetting facet) that are in contact with the TPM lobe are denoted in gray color, and the facets (α facet) that are adjacent to the wetting facets are denoted in yellow. The remaining facets (β facets) that are not adjacent to the wetting facet are denoted in green. For **S4**, there is an additional facet (γ facet) that is not adjacent to the wetting facet, which is denoted in red. The assembly of the particles only occurs when the β facets interact with the substrate, whereas the other cases result in no assembly due to the significant steric effect from the TPM lobe.

The detailed explanation and illustration are included in the revised manuscript in Fig. 5 and Fig. 6.

Reviewer 2:

vii) Why do the TPM droplets attached to the MOF particles survive the rather harsh centrifugation, pelleting and redispersion treatment?

Authors:

The TPM droplets are negatively charged, providing electrostatic repulsion to stabilize them in solution. The droplets are also slightly crosslinked due to the siloxane networks. They are generally more stable than common oil droplets. In addition, we added trace amount of F108 polymer (far below its CMC) to form polymer brush on the surface of TPM droplets, which

again prevents droplets from merging. In terms of centrifugation, we also used a low speed, 500 rpm (or approximate 20 g), making sure the centrifugation was as gentle as possible to avoid merging of the droplets. The droplets will indeed coalesce if much stronger centrifugation is used. Purification TPM oil-containing colloids by centrifugation have also been previously used by other researchers (*Nature* **550**, 234-238 (2017)).

Reviewer 2:

vii) Surface evolver simulation was certainly not established by these authors. References to earlier reports on the techniques are missing.

Authors:

Thanks for the suggestion. We have added the reference (*Exp. Math.* **1**, 141-165 (1992)). In fact, the original developer of Surface Evolver, prof. Kenneth Brakke, has helped us during the simulation process. We have acknowledged him in our paper.

Reviewer 2:

viii) Modelling of the degree of chirality; a schematic representation that shows the chosen geometric model would be helpful to readers.

Authors:

Thanks for the suggestion. The schematic representations of the chosen geometric models are added in Fig. 7 in the revised manuscript, which is shown below as Fig. R8.

Figure R8. Degree of chirality. **a**, Illustration of calculation for degree of chirality (DC). **b**, The achiral reference (hypothetic) of a trefoil trimer. The TPM lobe is represented as a gray

cycle and UiO-66 is represented as a light red triangle. The vertex for calculation is presented as a black dot. Six black dots are selected from a TPM sphere, and three black dots are picked from a UiO-66 triangle. **c-e**, Illustration of trefoil trimer of **S2 (c)**, **S3 (d)** and **S4 (e)** and the corresponding confocal overlay images. The center is represented as a red dot in the middle of each trimer. The numbers at the bottom are the *DC* values of each assembly. Scale bar: 3 μm for **(c-e)**.

Reviewer 2:

ix) The manuscript contains several typos that should be corrected.

Authors:

We thank the reviewer for catching the typos. We have now fixed them.

Reviewer 3:

The Nature Communications article by Zhang details the synthesis and self-assembly of Janus particles composed of polyhedral MOF and polymeric TPM using a method published by this group in 2022. This method allows for the precise control of the shape utilizing wetting/dewetting stimulated via Triton X-100. The particles presented exhibit reduced symmetries due in part to the polyhedral MOF used, which is unique to the field and possess the potential to use other similar MOF as a means to access other symmetries. The article explains the presence of only four distinct products after dewetting and thoughtfully used simulations to rationalize these distinct products. Thru the use of simulations, they have elucidated the step-wise asymmetric de-wetting which is entirely new since most Janus particle in literature experience a symmetric de-wetting process. This highlights the possibility that other polyhedron may utilize this same step-wise asymmetric de-wetting. The authors use a combination of shape and depletion forces to achieve the self-assemblies, this is a common practice and is well documented in the literature. The assemblies of Janus particles S2-4 result most notably in chiral clusters, along with others. The authors further defined the degree of chirality of the resultant assemblies. And presented a thorough and thoughtful discussion of the resultant assemblies.

Overall, the manuscript is well-written and easy to follow. The manuscript puts the created Janus particles presented into context with current state of the field, as well as builds a solid case that this method could work for other polyhedrons beyond the UiO-66 octahedral used here. Additionally, the authors presented a thorough and thoughtful discussion. The science is very interesting. Congrats to the authors. Therefore, after the minor corrections highlighted below I recommend publication of this article.

Authors:

We are delighted about the very positive feedback, describing our science as “very interesting”, and has recommended publication. We have clarified the points raised, as shown below, and we thank the reviewer for the comments.

Reviewer 3:

1. Duplicate word in sentence (line 36-38) of surprisingly.

Authors:

Thanks for your suggestion. We have revised it accordingly.

Reviewer 3:

2. The abstract would benefit from saying what type of polyhedral MOF is used (UiO-66), as well as specifying the polyhedral as an octahedral. Because this paper only discusses one polyhedral specifically the octahedral UiO-66 and has not proven any other polyhedral shapes to use the same step-wise pathway presented later.

Authors:

We have revised the abstract accordingly.

Reviewer 3:

3. A few sentences as an explanation on the difference in affinity between the TPM and UiO-66 to Triton X-100 would be beneficial.

Authors:

Thanks for your suggestion. We have added explanation in the revised manuscript.

The role of Triton X-100 (TX) is to decrease the surface tension between TPM and UiO-66 surface to the aqueous solution. Though it is hard to quantify, we speculate that UiO-66 tends to absorb more TX than TPM oil does. This would result in increased contact angle because the surface tension between UiO-66 and water decreasing much more rapidly than that between TPM oil and water. Similar dewetting phenomenon has been previously observed and utilized by us in other system (*J. Am. Chem. Soc.* **141**, 14853-14863 (2019)).

Reviewer 3:

4. Figure 2a would benefit from sub-headers i (top), ii (middle) and iii (bottom) for clarity in the context of the discussion.

Authors:

Thanks for your suggestion. We have added the sub-headers to the revised manuscript.

Reviewer 3:

5. Lines 155-157, Say that symmetric de-wetting is associated with a higher surface energy but the graph presented in Fig 2c shows the symmetric de-wetting having a lower energy? Is the key in Fig 2c reversed?

Authors:

We thank the reviewer for catching this error. We had mistakenly labeled the symmetric dewetting and asymmetric dewetting. The red circles (upper line) should refer to the symmetric dewetting and the blue squares (lower line) should refer to the asymmetric dewetting. We have corrected the labels accordingly.

Reviewer 3:

6. What informed the choice of depletant? size? charge? What happens if a different depletant is used? Do the assemblies become stereoselective?

Authors:

We thank the reviewer for these questions. The use of depletion interaction for assembling MOF particles have been reported by us previously (*Nat. Commun.* **13**, 3980 (2022)). For it to work, CTAC was used, which as an ionic amphiphile serve two purposes. First, it adsorbs on the MOF surface to increase the surface charge, increasing MOF's colloidal stability. Second, it forms micelles which act as the depletant.

While we continued to use the strategy herein—since it works, we think other depletants would also work if there are no practical issues. Of course, the size of the depletant needs to be considered to distinguish the shape difference of the two lobes of the Janus particle. In fact, we have tried the commonly used PEG polymer as depletant in buffer solution, but the polymer adsorbs on the surface of MOF particles and the buffer solution gradually degrade the particle.

Finally, using depletion force to realize stereoselective colloidal assembly is a bold idea we could explore in the future, possibly by using some chiral nanoparticles (*Nature* **601**, 366-373 (2022)). However, the outcome of which is hard to predict at this stage.

Reviewer 3:

7. Fig 3d, the inset is not necessarily adding to the discussion occurring in the rest of Fig 3, as evidenced by the lack of discussion in the text. Could be removed or moved to serve more of a purpose in later figures.

Authors:

Thanks for your suggestion. We have removed the inset images in Fig. 3d in the revised manuscript and moved it into the SI.

Reviewer 3:

8. The discussion of the antiparallely needs to be more flushed out or discarded. The associated Figure 5f is not adding value to the figure and could be moved to the SI.

Authors:

Thanks for your suggestion. We have removed the Fig 5f and moved it to the SI in the revised manuscript. We have also highlighted the point about antiparallel facet overlap.

Reviewer 3:

9. Fig 5g is not necessary to show the hexagonal superlattice because it's not adding to the discussion of the findings since it's an impossible lattice to achieve with the designed particles as noted Line 283-285.

Authors:

Thanks for your suggestions. In response, we have removed the illustration about hexagonal lattices.

Reviewer 3:

10. Fig 5h-j lacks the symmetry planes (dashed lines) presented in previous figures, adding them in would be beneficial reader clarity during the discussion.

Authors:

Thanks for your suggestion. We have added the symmetry planes (dashed lines) in Fig 5 in the revised manuscript.

Reviewer 3:

11. Line 283-285, states that you can only have 3 particles in a cluster but Fig 5j presents a tetramer cluster? So, can you only have max 3 in an assembly or do the TPM portions cause too much steric hinderance to form extended structures? These sentences just need some rewording so the authors aren't contradicting themselves.

Authors:

We thank the review for the comment and suggestions. We have revised the relevant part.

As we show in Fig. 5, the major assembly of **S3** is trimer clusters, including mostly the chiral trimers and Y-shape trimer. With a wide open, the Y-shape trimer cannot adopt another particle to form the tetramer with a small quantity. Through further experimentation, we have found that the tetramer has to do with the small fraction of **S2** particles in the **S3** sample (Fig. 1h). When we purposely mix **S2** sample with **S3** sample, we can drastically increase the yield of tetramer (Fig. R7), which supports our point. We note that in the synthesis, converting **S2** to **S3** is realized by addition of TX-100; insufficient amount of TX-100 would lead to a mixture of **S2** and **S3**.

Figure R9. **a, b**, Optical microscope images of assemblies of **S2** (**a**) and the mixture of **S2** and **S3** (**b**). **c**, Number percentage of trimer and tetramer in assemblies of **S3** and the mixture of **S2** and **S3**. Scale bars: 25 μm for (**a, b**).

Therefore, the formation of tetramer is the result of co-assembly between **S3** and **S2** particles. In the current paper, we focus on the assembly structure of Janus particles in pure state rather than the co-assembly of different states. That's why we claim the maximum number of contact faces for **S3** assemblies is 3 instead of 4.

These explanations have been included in SI in the revised manuscript.

Reviewer 3:

12. Figure 6 is the only assembly figure to not present a number percentage of each species that forms, please add.

Authors:

Thanks for your suggestion. We have added the number percentage in Figure 6 in the revised manuscript.

Reviewer 4:

In this study, Wang et al. present a novel category of particles called Janus particles, which exhibit controllable patch and symmetries through synthetic means. The method involves incorporating a polyhedral MOF particle within a polymerizable oil matrix. The extent of encapsulation is adjusted by controlling the concentration of surfactants and the dewetting process, which determines final shape of the Janus particle. The authors induced self-assembly of these Janus particles by employing depletion forces. Intriguingly, they also observed the formation of chiral colloid assemblies using non-chiral building blocks. This study effectively combines the realms of patchy particles and the growing utilization of MOF particles for self-assembly. The authors demonstrate remarkable control over the fabrication of Janus particles and the formation of diverse colloidal clusters, which depend on the symmetry of the Janus particle. I find this study highly compelling, well explained and believe it is well-suited for publication in Nature Communications. However, I do have a few comments for the authors:

Authors:

We thank the reviewer for examining our work and for the valuable comments. We are happy that the reviewer considered our idea “compelling”, “well-explained”, and “suitable for publication”.

Reviewer 4:

1. The authors should introduce and illustrate with examples the concepts of colloidal MOFs and self-assembly of colloidal MOFs in the introduction. Both concepts, basic for this study, are barely touched in the introduction. The authors should see some recent reviews published in Chem. Soc. Rev.

Authors:

Thanks for your suggestion. We have introduced more background and information about the concepts of colloidal MOF and its self-assembly in the revised manuscript. The recent review published have been cited (*Chem. Soc. Rev.* **52**, 2528-2543 (2023)).

Reviewer 4:

2. It would be beneficial to include more representative SEM images of the different Janus particles on a larger scale.

Authors:

Thanks for your suggestion. We have put the SEM images on a larger scale in SI in the revised manuscript. They are also shown here for easy reference.

Figure R10. Morphology of Janus particles on large scale. a-d, SEM images of S1 (a), S2 (b), S3 (c) and S4 (d) in a large scale. Scale bars: 10 μ m.

Reviewer 4:

3. It is unclear why the authors did not present any SEM images of the assembled clusters and whether they are stable only in wet conditions. Clarification on this point would be helpful.

Authors:

We thank the reviewer for the suggestion. We had tried to dry the sample (using various methods including freeze drying) for SEM characterization but could not maintain its assembly structure. This could be that the clusters are held together by depletion force (in wet condition), which is relative weak. The assemblies fell apart or collapsed upon drying. Since optical microscope (bright field and confocal) have provided clear information, we kindly request not to provide SEM image in the current manuscript. In the long run, we will explore crosslinking strategies to covalently fix the assembly.

Reviewer 4:

4. Have the authors explored Monte Carlo simulations to support their calculations on the various cluster assemblies? The incorporation of this type of simulations could strengthen their findings.

Authors:

We thank the reviewer for the comment and suggestion. We have conducted Monte Carlo simulation to model the depletion interaction and particle assembly of our Janus particles. We have successfully replicated the chiral clusters, of various degree of chirality, whose trend agrees with and supports our calculation based on experiments. The results are included in the main text and SI and are shown below in detail.

Figure R11. Monte Carlo simulation of self-assembly. **a** Selective assembly of UiO-66 particles and TPM particles in different settings of fugacity (f). **(b)** Self-assembly of S4 when $f = 350$. **(c)** Clusters of S4 by simulation. **d**, Illustration of the definition of r . **e**, Self-assembly of S2 with different patch ratios (r) when $f = 350$. The differences between the assembled clusters can be revealed by the size difference of the triangle (orange) in the center of the cluster, as a result of the different arrangement of the octahedral particles (blue). **f**, The degree of chirality (DC) of S2 trimer with different values of r .

We perform Monte Carlo simulation (using HOOMD-blue) to explore the assembly of Janus particles. The implicit depletant simulation algorithm, previously developed by Glotzer, et al., is used to simulate depletion interaction between anisotropic colloids in an implicit way. In this algorithm, one can control the depletant size and depletant fugacity to change the depletion force between large particles. The fugacity, f , is defined as $f = e^{\beta\mu_p} / \lambda_p^3$, where μ_p denotes the chemical potential of the depletants, β denotes $1/k_B T$ (reciprocal of the product of the Boltzmann constant k_B and the temperature T), and λ_p denotes the de Broglie wavelength. The fugacity is a function with positive correlation to the number density of depletants, whereby a higher value of f is associated with a stronger depletion interaction (J. Chem. Phys. 143, 184110 (2015)).

Figure R12. Monte Carlo simulation of S3 assembly. a, Self-assembly of S3 when $f=350$. **b,** Clusters of S3 by simulation.

To verify the model, we first simulate the selective assembly of octahedral particles in the presence of spheres. The ratio between the radius of the sphere and the size of octahedron (φ) is set as 0.7, and the ratio between the size of sphere and the size of depletants is 20. All particles are confined to move only on the bottom wall of a 3D box, mimicking the behavior of particles on a substrate in experiments. For octahedral particles, one facet is set to be parallel to the bottom. The particles are allowed to translate only in the xy-plane and rotate only around the z-axis. We have determined that when $f = 250$ to 400, the octahedral particles can assemble, while the sphere do not, due to weaker interactions (Fig. R11a, R13a).

We then simulate the assembly of Janus particles. The S4 particle is first tested, built in simulation by connecting a sphere to an octahedron. The particles are confined to the substrate with a predetermined orientation (i.e., with one facet of the octahedral parallel to the substrate). Our simulation shows that chiral clusters can emerge when $f = 350$, consistent with the experimental results (Fig. R11b). The chiral clusters have both handedness. Various other clusters of S4 can also be reproduced successfully from simulation, including the dimer, achiral trimer. (Fig. R11c). The simulation of S3 is similar to that of S4 (Fig. R12).

Figure R13. a, Selective assembly of UiO-66 particles and TPM particles in different settings of fugacity (f). **b**, Self-assembly of S2 with different patch ratios (r) when $f = 350$.

We next simulate other configurations of Janus particles, S2, where the steric of the TPM lobe have shown to play a role in the assembly. The particle model is built by overlapping a sphere with an octahedron according to the actual particle shape, where four facets have been covered and other four facets in a pyramid are exposed. The size of the sphere is varied to realized different patch ratios of the Janus particles.

To evaluate the influence of the TPM lobe on the degree of chirality (DC) of the assembly, we perform simulations with different patch ratios ranging from 0.66 to 0.54, by varying the size of the sphere in the model. While we have obtained trimer clusters, the DC values decrease as the patch ratio decreases. This result is consistent with our experimental observation, wherein S2 trimers exhibit obvious chirality when $r = 0.81$ and small chirality when $r = 0.64$; few assemblies are observed when $r = 0.44$ (Fig. R4). The variation of DC in S2 clusters is attributed to the different steric effects exerted by the TPM lobes. The large TPM lobes impose strong steric effects that prevented the MOF patches from maximizing their contact area, resulting in low DC values, while the small TPM lobes have relatively weak steric effects that allow the

MOF patches to maximize their contact area, leading to high DC values. This result confirms that the chirality of the assembly could be altered by changing the size of the steric lobe.

REVIEWERS' COMMENTS

Reviewer #1 (Remarks to the Author):

The authors have addressed the questions and concerns that I raised in my initial review of this submission. With the introduction of MOF-hybrid colloids and their depletion-induced self-assembly, the work represents an interesting addition to the literature. Therefore, I recommend publication in Nature Communications.

Reviewer #2 (Remarks to the Author):

The authors made comprehensive revisions with regard to all points and queries either raised by me or the other reviewers. This even includes important new data that was added to clarify some decisive aspects. I can now recommend publication of the revised manuscript without any reservations.

Reviewer #4 (Remarks to the Author):

The authors have properly addressed all my comments and concerns raised on the original submission, so I support its acceptance in Nature Communications.